# Stay Unique, Stay Efficient: Preserving Model Personality in Multi-Task Merging

## Abstract

Model merging has emerged as a promising approach for enabling multi-task capabilities without additional training. However, existing methods often suffer from substantial performance degradation compared to individual models, even on similar tasks, highlighting the importance of preserving task-specific information. This paper introduces an approximation-based personalized merging method, **D**ecomposition, **T**hresholding, and **S**caling (**DTS**), which retains task-specific information with minimal storage overhead. DTS first performs singular value decomposition on the task-specific information and preserves only a small subset of singular values and vectors. It then applies a novel thresholding strategy to group the elements within each singular vector and computes a scaling factor for each group. To further support generalization to unseen tasks, this paper extends DTS with a variant that leverages the semantic similarity of task characteristics to merge task-specific information in a data-free manner. Extensive experiments demonstrate that DTS consistently outperforms state-of-the-art baselines, delivering superior performance with just 1% extra storage per task. Furthermore, experiments on unseen tasks show that the DTS variant achieves significantly better generalization performance. Our code is available in the **supplementary materials**.

## 1 Introduction

With the thriving of pre-trained models and the growth of open-source ecosystems such as Huggingface (Wolf et al., 2019) and timm (Imambi et al., 2021), fine-tuning has become the standard approach for adapting general-purpose models to diverse downstream tasks (Devlin et al., 2019; Dodge et al., 2020; Bommasani et al., 2021; Paul & Chen, 2022; Ye et al., 2023). This trend has resulted in a proliferation of fine-tuned models across domains. However, maintaining and deploying separate fine-tuned models for each task incurs substantial memory and infrastructure costs, rendering it impractical in resource-constrained environments. Multi-task learning mitigates this issue by training a single model jointly on multiple tasks (Caruana, 1997; Zhang & Yang, 2018; 2021), but it requires simultaneous access to all training data and often entails high computational costs.

Model merging (Ainsworth et al., 2023; Zhao et al., 2025; Li et al., 2025) has recently emerged as an appealing alternative. By combining the parameters of multiple fine-tuned models into one, it enables efficient knowledge fusion without the need for original training data or further optimization. Among existing approaches, Task-Arithmetic (Ilharco et al., 2023) interpolates parameters linearly but suffers from performance degradation due to parameter conflicts (Yadav et al., 2023; Qi et al., 2025). Ties-Merging (Yadav et al., 2023) alleviates this by selectively resetting minimally changed parameters and resolving sign conflicts. AdaMerging (Yang et al., 2024) further refines merging weights using additional training data. Despite these improvements, a significant gap remains between merged and individually fine-tuned models.

In this paper, we study performance degradation in model merging through the lens of task similarity. Specifically, we conduct pairwise model merging by combining the model fine-tuned on the SVHN (Netzer et al., 2011) dataset with models fine-tuned on each of the other seven datasets in the visual benchmark (see Sec. 4.1 for details). Fig. 1(a) shows the best and worst performance of the merged models when evaluated on SVHN, corresponding to merging with the most similar task (MNIST (Deng, 2012)) and the most dissimilar task, respectively. Surprisingly, we observe significant performance drops even when merging models trained on similar tasks.

A comparable trend emerges in natural language processing, as illustrated in Fig. 1(b). These results underscore the persistent nature of parameter conflicts across tasks and emphasize the importance of preserving task-specific information in model merging.

Motivated by these observations, we propose **DTS**, an approximation-based personalized model merging method that leverages **D**ecomposition, **T**hresholding, and **S**caling to efficiently preserve task-specific information while minimizing storage overhead. DTS applies singular value decomposition (SVD) (Lange & Lange,

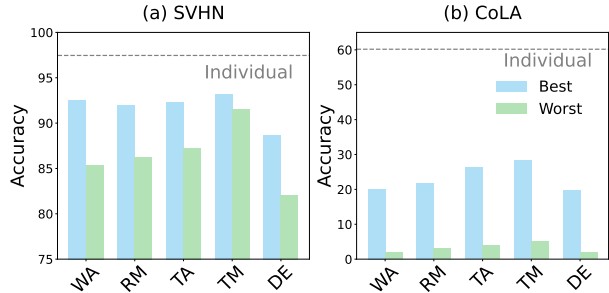

Figure 1: Performance of the pairwise merged model: we pairwise merge models fine-tuned on SVHN or CoLA with those fine-tuned on other seven datasets, reporting the best and worst performance across these combinations. WA, RM, TA, TM, and DE refer to Weight-Averaging, RegMean, Task-Arithmetic, Ties-Merging, and DARE, respectively.

2010) to each parameter matrix in the task-specific information, retaining only a small subset of the filtered singular values and vectors. This selective retention significantly reduces storage requirements while maintaining essential task information. To further reduce storage costs, we introduce a novel thresholding strategy to approximate the singular vectors. Unlike prior work (Qi et al., 2025), which retains a subset of elements using a simple binary mask, we argue that this coarse masking may lead to the loss of fine-grained information. Instead, we propose a novel thresholding strategy that thresholds the elements of each singular vector into four groups and assigns a scaling factor for each group, as illustrated in Fig. 3. This strategy improves the approximation for the original task-specific information. In addition to applying DTS to task vectors (DTS-T), we introduce a new form of task-specific information termed the difference vector, which captures the parameter difference between individually fine-tuned and merged models. We apply DTS to this difference vector, referring to it as DTS-D.

To enable generalization to unseen tasks, we propose a variant of DTS that adaptively integrates task-specific information from seen tasks. Unlike prior methods that require additional data or trainable routers to determine merging weights, our approach computes weights based on the average semantic similarity between seen and unseen task characteristics. This design is entirely data-free, offering both efficiency and scalability. Experimental results demonstrate that DTS consistently outperforms state-of-the-art baselines on standard multi-task model merging benchmarks, achieving strong performance with only 1% extra storage per task. Furthermore, on unseen tasks, the DTS variant delivers superior generalization performance compared to all competing methods. Our contributions can be summarized as follows:

- This paper finds that merging models from similar tasks still results in performance degradation, highlighting the importance of preserving task-specific information. To address this challenge, we propose **DTS**, a lightweight yet effective approximation-based personalized merging method.
- We propose two methods, DTS-T and DTS-D, which apply DTS on task vectors and difference vectors, respectively. Additionally, we present a variant of DTS that fuses task-specific information based on the semantic similarity of task characteristics to support generalization to unseen tasks.
- Experimental results on standard multi-task model merging benchmarks demonstrate that our method consistently outperforms state-of-the-art baselines, while requiring only 1% extra storage per task. Furthermore, experiments on unseen tasks show that the DTS variant achieves superior generalization performance compared to all baseline methods, highlighting its effectiveness in both seen and unseen task settings.

## 2 RELATED WORK

Model merging (Ainsworth et al., 2023; Ilharco et al., 2023; Touvron et al., 2023), which aims to fuse multiple fine-tuned models into a single comprehensive model, has attracted increasing attention with the release of numerous publicly available model checkpoints (Wolf et al., 2019; Dodge et al., 2020; Jordan et al., 2023). Model merging significantly reduces storage and deployment cost by unifying multiple models into a single one, without additional training (Yu et al., 2024; He, 2024; Liu

et al., 2025; Sun et al., 2025). Depending on whether the merged model is static or tailored per task, existing methods can be categorized into basic model merging and personalized model merging.

## 2.1 BASIC MODEL MERGING

Basic model merging (Ilharco et al., 2023; Wortsman et al., 2022; Jin et al., 2023; Matena & Raffel, 2022; Yadav et al., 2023; Yu et al., 2024; Davari & Belilovsky, 2024) focuses on universal strategies for merging fine-tuned models into a single shared model without introducing additional memory overhead. A canonical example is Weight-Averaging (Wortsman et al., 2022), which simply averages parameters from different tasks. Task-Arithmetic (Ilharco et al., 2023) refines the merging process by introducing task vectors, proposing that simple arithmetic operations on these vectors can effectively modify models and yield a better merged model. Building on this idea, DARE (Yu et al., 2024) and Ties-Merging (Yadav et al., 2023) propose pruning and scaling task vectors, assuming that not all parameters contribute equally. However, these techniques depend on manually tuned merging coefficients. AdaMerging (Yang et al., 2024) introduces adaptive learning to automate coefficient selection but incurs additional training cost. Other approaches like Fisher-Merging (Jin et al., 2023) and RegMean (Matena & Raffel, 2022) compute merging coefficients via Fisher information or inner product matrices, but require costly gradient computations and are prone to instability. Basic model merging across different tasks often results in parameter conflicts and a loss of task-specific information, leading to a substantial performance gap compared to individual models (Lee et al., 2025; Stoica et al., 2025). To this end, we explore the personalized model merging in this work.

## 2.2 PERSONALIZED MODEL MERGING

Personalized model merging (Muqeeth et al., 2023; Wang et al., 2024; Lu et al., 2024; Tang et al., 2024; Huang et al., 2024; Zhu et al., 2025) enhances the merged model with task-specific components to boost performance across a variety of tasks. SMEAR (Muqeeth et al., 2023) and Twin-Merging (Lu et al., 2024) store full model parameters for each task and employ routing mechanisms to perform weighted parameter fusion based on expert distributions. WEMOE (Tang et al., 2024) introduces test-time adaptation by merging most weights while converting MLP layers into a mixture-of-experts (MoE) module. DaWin (Oh et al., 2025) similarly retains full models and uses entropy over unlabeled test samples to assess task relevance. 1bit-Merging (Liu et al., 2025) stores task-specific attention or MLP modules and uses a trained router to combine them dynamically. While effective, these methods typically require access to training data or incur high storage costs. In contrast, EMR-Merging (Tang et al., 2024) selects a unified base model and generates lightweight, data-free modulators for each task. T-Switch (Qi et al., 2025) further reduces memory by storing task vectors in a binarized form. Nonetheless, existing personalized methods often suffer from either data dependency or excessive storage demands. In this work, we aim to retain essential task-specific information with minimal memory overhead, mitigating parameter conflicts without relying on additional data or training.

## 3 METHOD

### 3.1 PROBLEM FORMULATION

In this paper, following the setup of prior model merging works (Ilharco et al., 2023; Yadav et al., 2023; Yu et al., 2024; Qi et al., 2025), we consider a scenario involving $N$ tasks with corresponding datasets $\{\mathcal{D}_n\}_{n=1}^N$, where each sample $(\boldsymbol{x}_n, y_n) \in \mathcal{D}_n$ belongs to the $n$-th task. Let $f_{\boldsymbol{\theta}_0}$ denote a pretrained model with parameters $\boldsymbol{\theta}_0 \in \boldsymbol{\Theta}$, and let $f_{\boldsymbol{\theta}_1}, \ldots, f_{\boldsymbol{\theta}_N}$ represent task-specific models fine-tuned on each $\mathcal{D}_n$, where $\boldsymbol{\theta}_n$ is the fine-tuned weights for the $n$-th task. Following Task-Arithmetic (Ilharco et al., 2023), the task vector $\boldsymbol{\tau}_n$ for the $n$-th task is defined as $\boldsymbol{\tau}_n = \boldsymbol{\theta}_n - \boldsymbol{\theta}_0$, which serves as a common form of task-specific information. In addition, we introduce a new form of task-specific information termed the **difference vector**, defined as $\boldsymbol{d}_n = \boldsymbol{\theta}_n - \boldsymbol{\theta}_m$, where $\boldsymbol{\theta}_m$ is the parameter set obtained via a base merging method such as Ties-Merging (Yadav et al., 2023). As an illustrative example, we consider task vectors as task-specific information. The goal of model merging is to combine the set $\{\boldsymbol{\tau}_n\}_{n=1}^N$ with the pre-trained model to produce a merged model that performs well on the union of all task datasets, $\mathcal{D} = \bigcup_{n=1}^N \mathcal{D}_n$, formulated as:

$$\min \mathbb{E}_{(\boldsymbol{x},y)\in\mathcal{D}} \ell\left(f_{\boldsymbol{\theta_m}}(\boldsymbol{x}), y\right), \text{ where } \boldsymbol{\theta_m} = \mathcal{M}(\boldsymbol{\theta}_0, \{\boldsymbol{\tau}_n\}_{n=1}^N). \tag{1}$$

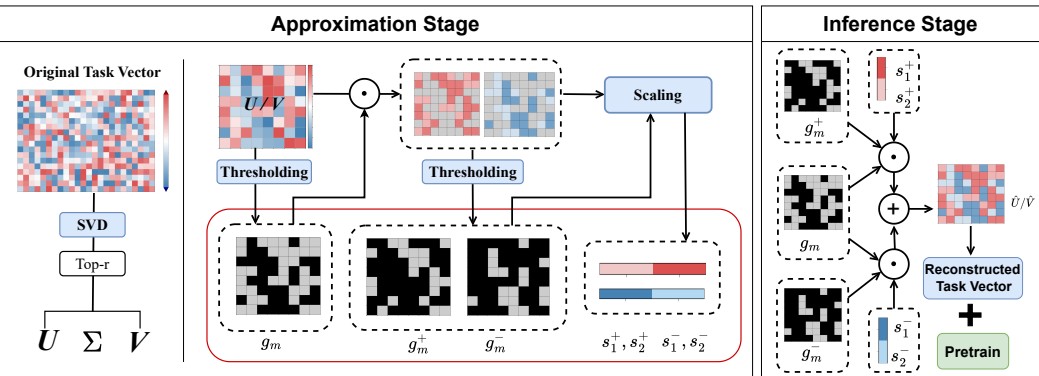

Figure 2: Framework overview. In the approximation stage, we first apply singular value decomposition to the task-specific information. Next, a novel thresholding strategy is used to group the elements of each singular vector, followed by computing a scaling factor for each group. During inference, the task-specific information is reconstructed using the approximated singular vectors.

Here, $\mathcal{M}$ denotes a general merging function. For instance, in Task Arithmetic (Ilharco et al., 2023), it takes the form $\mathcal{M}(\boldsymbol{\theta}_0, \{\boldsymbol{\tau}_n\}_{n=1}^N) = \boldsymbol{\theta}_0 + \sum_{n=1}^N \gamma_n \boldsymbol{\tau}_n$. More advanced personalized strategies, such as generating task-specific modulators (Huang et al., 2024), can also be employed to merge models across diverse tasks.

## 3.2 Decomposition, Thresholding, and Scaling

Previous studies (Zhou et al., 2024; Marczak et al., 2025; Yan et al., 2025) have shown that merged models often suffer notable performance degradation compared to their individually fine-tuned counterparts, primarily due to parameter conflicts between fine-tuned models. In this work, we examine this issue through the lens of task similarity. Specifically, we perform pairwise model merging by combining a model fine-tuned on SVHN (Netzer et al., 2011) with models fine-tuned on each of the other seven datasets in the benchmark (see Sec. 4.1 for details).

Fig. 1(a) shows the best and worst SVHN performance among the merged models, corresponding to merging with the most similar task (MNIST (Deng, 2012)) and the most dissimilar one, respectively. Surprisingly, substantial performance drops are observed even when merging models from similar tasks. A similar pattern emerges in natural language processing tasks, as illustrated in Fig. 1(b). These findings suggest that parameter conflicts are inherent, even across semantically related tasks, and underscore the importance of preserving personalized information to maintain model performance. This insight motivates our objective: to retain complete task-specific information while minimizing additional storage overhead.

In this paper, we propose to directly store the task-specific information of individual models. However, storing full model parameters leads to substantial storage overhead, which is impractical in resource-constrained settings. To solve this, we introduce **D**ecomposition, **T**hresholding, and **S**caling (**DTS**), a personalized method designed to approximate task-specific information while preserving its effectiveness, as illustrated in Fig. 2. To better illustrate our approach, we use the task vector as an example of task-specific information. Specifically, we apply singular value decomposition to each task vector and retain only the top-$r$ singular values, as follows:

$$\boldsymbol{U}_n, \boldsymbol{\Sigma}_n, \boldsymbol{V}_n = \text{SVD}_r(\boldsymbol{\tau}_n), \tag{2}$$

where $\boldsymbol{U}_n$ and $\boldsymbol{V}_n$ denote the left and right singular vector matrices, respectively, and $\boldsymbol{\Sigma}_n$ contains the singular values. Here, r is the proportion of singular values retained.

To further reduce storage, we threshold the decomposed components. Unlike previous work (Qi et al., 2025), which uses a simple binary mask to retain a subset of elements, we argue that this coarse masking approach may result in the loss of fine-grained information. Instead, we propose a novel thresholding strategy that thresholds the elements of each singular vector into four groups and computes a scaling factor for each group. Specifically, taking the left singular vector matrix $\boldsymbol{U}_n$ as an

example, we first mark the sign of each element as follows:

$$g(U_{n,j}) = \begin{cases} 1, & \text{if } U_{n,j} > 0, \\ 0, & \text{otherwise,} \end{cases} \tag{3}$$

where $U_{n,j}$ is the $j$-th element of $\boldsymbol{U}_n$. This thresholding function encodes the sign of each parameter using only 1 bit, dividing the parameters into positive and negative groups. To preserve more fine-grained information, we further threshold the positive and negative values into two subgroups, respectively. Taking the positive values as an example:

$$g^+(U_{n,j}) = \begin{cases} 1, & \text{if } U_{n,j} > \lambda, \\ 0, & \text{otherwise,} \end{cases} \tag{4}$$

where $\lambda$ is the median of the positive entries in $\boldsymbol{U}_n$. This extended thresholding partitions positive elements into "large" and "small" groups, as illustrated in Fig. 3. The same strategy is applied to the negative values. During the inference stage, we reconstruct the positive portion of $\boldsymbol{U}_n$ as:

$$\hat{\boldsymbol{U}}_n^+ = s_1^+ \cdot g(\boldsymbol{U}_n) \odot g^+(\boldsymbol{U}_n) + s_2^+ \cdot g(\boldsymbol{U}_n) \odot (1 - g^+(\boldsymbol{U}_n)), \tag{5}$$

where $s_1^+$ and $s_2^+$ are scaling factors used to align the magnitudes of the approximated and original task vectors. These are computed as:

$$s_1^+ = \frac{\|\boldsymbol{U}_n \odot g(\boldsymbol{U}_n) \odot g^+(\boldsymbol{U}_n)\|_2}{\|g(\boldsymbol{U}_n) \odot g^+(\boldsymbol{U}_n)\|_2}, \quad s_2^+ = \frac{\|\boldsymbol{U}_n \odot g(\boldsymbol{U}_n) \odot (1 - g^+(\boldsymbol{U}_n))\|_2}{\|g(\boldsymbol{U}_n) \odot (1 - g^+(\boldsymbol{U}_n))\|_2}. \tag{6}$$

In Fig. 3, the computed values of $s_1^+$ and $s_2^+$ are 8.00 and 3.14, respectively. The same reconstruction procedure is applied to the negative values of $\boldsymbol{U}_n$. We apply the same method to obtain the approximated $\hat{\boldsymbol{V}}_n$, and more implementation details are provided in Appendix A.

For one-dimensional components in the task vector, we bypass the SVD step and directly apply thresholding and scaling. Importantly, our method requires storing only three masks per $\boldsymbol{U}$ or $\boldsymbol{V}$ matrix to approximate the original task-specific information. Each mask element occupies just 1 bit,

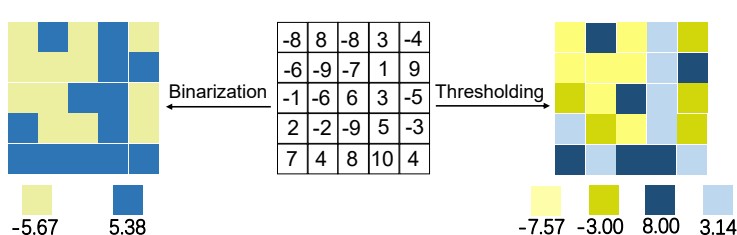

Figure 3: A toy example of our thresholding strategy.

leading to a $32\times$ reduction in storage compared to full-precision parameters. This significantly lowers storage costs while preserving essential task-specific structure.

Finally, we reconstruct the task-specific parameters dynamically for each task $\mathcal{D}_n$ as:

$$\hat{\boldsymbol{\theta}}_n = \boldsymbol{\theta}_0 + \hat{\boldsymbol{U}}_n \times \boldsymbol{\Sigma}_n \times \hat{\boldsymbol{V}}_n. \tag{7}$$

This reconstruction preserves task-specific information while entirely avoiding parameter conflicts across tasks. By applying DTS to task vectors (DTS-T) or difference vectors (DTS-D), our method requires only 1% of the storage of a full-precision model. This makes DTS particularly well-suited for memory-constrained environments.

### 3.3 EXTENDING DTS FOR UNSEEN TASKS

Although previous DTS methods applied to task vectors effectively mitigate parameter conflicts, they are not directly applicable to unseen tasks. While $\boldsymbol{\theta}_m$ can serve as a generic model for such tasks, it overlooks task similarity, which may hinder generalization performance. In scenarios where unseen tasks are semantically similar to seen ones, incorporating weighted task-specific information can improve transferability. To this end, we introduce a variant of DTS. For clarity, we illustrate the approach using DTS-D and the approximation-based personalized components $\tilde{\mathbf{d}}_n$. Unlike existing methods that rely on additional training data or learnable routers to compute merging weights, our method leverages the semantic similarity of task characteristics, avoiding extra computational and storage costs. Specifically, taking classification tasks as an example, where class names can serve as

Table 1: Multi-task performance (%) when merging ViT-B/32 models on eight tasks. ADR refers to the Accuracy Drop Rate, and AMR denotes the Additional Memory Rate. The best result is highlighted in **bold**, and the second-best result is underlined.

| Method | SUN397 | Cars | RESISC45 | EuroSAT | SVHN | GTSRB | MNIST | DTD | Avg. ↑ | ADR ↓ | AMR ↓ |
|---|---|---|---|---|---|---|---|---|---|---|---|
| Individual | 74.49 | 77.73 | 98.22 | 99.80 | 97.46 | 98.73 | 99.69 | 79.36 | 90.69 | – | – |
| Weight-Averaging (Wortsman et al., 2022) | 65.35 | 63.41 | 71.42 | 71.69 | 64.20 | 52.82 | 87.56 | 50.18 | 65.83 | 27.42 | 0 |
| Fisher-Merging (Matena & Raffel, 2022) | 68.69 | 69.21 | 70.73 | 66.41 | 72.91 | 51.17 | 87.94 | 59.99 | 68.38 | 24.61 | 0 |
| RegMean (Jin et al., 2023) | 65.35 | 63.53 | 75.61 | 78.66 | 78.10 | 67.49 | 93.75 | 52.02 | 71.81 | 20.71 | 0 |
| Task-Arithmetic (Ilharco et al., 2023) | 54.78 | 54.98 | 67.69 | 78.70 | 80.21 | 69.68 | 97.34 | 50.37 | 69.22 | 23.68 | 0 |
| Ties-Merging (Yadav et al., 2023) | 64.17 | 64.43 | 76.31 | 76.62 | 81.28 | 69.37 | 96.53 | 54.52 | 72.90 | 19.62 | 0 |
| DARE (Yu et al., 2024) | 64.76 | 63.08 | 71.02 | 70.70 | 62.04 | 50.68 | 86.17 | 50.64 | 64.89 | 28.45 | 0 |
| AdaMerging (Yang et al., 2024) | 64.44 | 68.05 | 79.31 | 93.80 | 87.06 | 91.93 | 97.56 | 59.11 | 80.16 | 11.62 | 0 |
| AdaMerging++ (Yang et al., 2024) | 66.61 | 68.34 | 82.28 | 94.11 | 89.54 | 89.01 | 98.18 | 60.66 | 81.09 | 10.58 | 0 |
| SVD | 71.18 | 71.60 | 96.95 | 99.64 | 97.24 | 98.02 | 99.66 | 77.58 | 88.98 | 1.88 | 5.87 |
| Twin-Merging (Lu et al., 2024) | 71.56 | 68.78 | 89.97 | 72.11 | 96.65 | 93.35 | 99.66 | 72.50 | 83.07 | 8.41 | 100.0 |
| WEMOE (Tang et al., 2024) | 73.92 | 77.36 | 93.58 | 99.11 | 96.25 | 98.64 | 99.57 | 76.43 | 89.36 | 1.47 | 49.95 |
| EMR-Merging (Huang et al., 2024) | 71.02 | 72.75 | 93.49 | 99.24 | 96.86 | 98.12 | 99.58 | 74.36 | 88.18 | 2.77 | 15.62 |
| TALL-Mask (Wang et al., 2024) | 73.02 | 77.38 | 97.63 | 99.38 | 97.15 | 98.46 | 99.66 | 77.61 | 90.04 | 0.71 | 15.62 |
| T-Switch (Qi et al., 2025) | 74.05 | 77.32 | 96.47 | 99.52 | 97.33 | 98.41 | 99.56 | 78.55 | 90.15 | 0.59 | 6.25 |
| DTS-T | 74.15 | 76.85 | 97.92 | 99.66 | 97.00 | 98.34 | 99.63 | 79.03 | 90.32 | 0.40 | 3.68 |
| DTS-D | 74.15 | 76.87 | 97.98 | 99.78 | 97.36 | 98.63 | 99.67 | 78.78 | 90.40 | 0.39 | 3.68 |
| DTS-T* | 74.01 | 76.13 | 97.74 | 99.52 | 96.96 | 98.15 | 99.60 | 78.51 | 90.08 | 0.67 | 0.98 |
| DTS-D* | 73.97 | 76.42 | 97.58 | 99.56 | 97.13 | 98.58 | 99.47 | 78.63 | 90.17 | 0.57 | 0.98 |

task-specific characteristics, we encode them with a text encoder to generate embeddings that capture the semantic relationships between tasks. For an unseen task, we compute the mean embedding $E_u$ over its class names. For seen tasks with embeddings $\{E_n\}_{n=1}^{N}$, we perform merging as follows:

$$\hat{\boldsymbol{\theta}}_u = \boldsymbol{\theta}_m + \sum_{n=1}^{N} \gamma_n \cdot \hat{\boldsymbol{d}}_n, \quad \gamma_n = \frac{\cos\_\text{sim}(E_u, E_n)}{\sum_{k=1}^{N} \cos\_\text{sim}(E_u, E_k)}, \tag{8}$$

where $\cos\_\text{sim}(\cdot, \cdot)$ denotes cosine similarity. The same strategy can be applied to DTS-T by substituting $\hat{\boldsymbol{d}}_n$ with $\hat{\boldsymbol{\tau}}_n$. For generative tasks, we treat task descriptions as task-specific characteristics and calculate the semantic similarity between tasks. The variant of DTS is entirely data-free and does not require access to training samples from unseen tasks, offering both flexibility and efficiency.

## 4 EXPERIMENTS

In this section, we evaluate the effectiveness and efficiency of the proposed method through conventional multi-task model merging experiments, as well as its ability to generalize to unseen tasks.

### 4.1 CONVENTIONAL MULTI-TASK MODEL MERGING

**Baselines.** We compare our method against both basic and personalized model merging methods. Basic methods include Weight-Averaging (Wortsman et al., 2022), Fisher-Merging (Matena & Raffel, 2022), RegMean (Jin et al., 2023), Task-Arithmetic (Ilharco et al., 2023), Ties-Merging (Yadav et al., 2023), DARE (Yu et al., 2024), and AdaMerging (Yang et al., 2024), all of which yield a single merged model without storing task-specific information. Moreover, personalized approaches retain additional task-specific parameters and include simple SVD, Twin-Merging (Lu et al., 2024), WEMOE (Tang et al., 2024), EMR-Merging (Huang et al., 2024), TALL-Mask (Wang et al., 2024), and T-Switch (Qi et al., 2025). Unless otherwise stated, we follow the configuration in T-Switch (Qi et al., 2025) for all baselines. We report results for both **DTS-T** and **DTS-D** using a default sparsity coefficient of $r = 0.3$. To further assess the efficiency of our approach under strict storage constraints, we also present results for **DTS-T\*** and **DTS-D\***, where $r$ is adaptively adjusted to ensure that the additional storage overhead remains below 1% across all backbones. More details about baseline methods are provided in Appendix B.1.

**Backbones and datasets**. We conduct experiments on visual classification, natural language processing, and natural language generation tasks. For visual classification, we use two variants of CLIP (Radford et al., 2021) vision encoders—ViT-B/32 and ViT-L/14. Following prior work (Lu et al., 2024; Qi et al., 2025), we evaluate on eight visual datasets: SUN397 (Xiao et al., 2010), Cars (Krause et al., 2013), RESISC45 (Cheng et al., 2017), EuroSAT (Helber et al., 2019), SVHN (Netzer et al., 2011), GTSRB (Stallkamp et al., 2011), MNIST (Deng, 2012), and DTD (Cimpoi et al., 2014). For natural language processing, we adopt RoBERTa (Liu et al., 2019) and GPT-2 (Radford et al., 2019) as backbones, and evaluate on eight tasks from the GLUE benchmark (Wang et al., 2018): CoLA (Warstadt et al., 2019), SST-2 (Socher et al., 2013), MRPC (Dolan & Brockett, 2005), STS-B (Cer et al., 2017), QQP (Chen et al., 2018), MNLI (Williams et al., 2018), QNLI (Rajpurkar et al.,

Table 2: Multi-task performance (%) when merging RoBERTa models on eight tasks. ADR refers to the Accuracy Drop Rate, and AMR denotes the Additional Memory Rate. The best result is highlighted in **bold**, and the second-best result is underlined.

| Method | CoLA | SST2 | MRPC | STSB | QQP | MNLI | QNLI | RTE | Avg. ↑ | ADR ↓ | AMR ↓ |
|---|---|---|---|---|---|---|---|---|---|---|---|
| Individual | 60.18 | 94.04 | 89.22 | 90.63 | 91.41 | 87.20 | 92.71 | 79.06 | 85.56 | – | – |
| Weight-Averaging (Wortsman et al., 2022) | 13.96 | 64.11 | 69.36 | 31.84 | 75.36 | 42.19 | 58.70 | 55.23 | 51.34 | 40.00 | 0 |
| RegMean (Jin et al., 2023) | 36.67 | 90.60 | 75.74 | 62.68 | 83.55 | 70.02 | 82.35 | 58.48 | 70.01 | 18.17 | 0 |
| Task-Arithmetic (Ilharco et al., 2023) | 18.78 | 85.89 | 79.90 | 74.03 | 83.78 | 59.08 | 69.67 | 62.09 | 66.65 | 22.10 | 0 |
| Ties-Merging (Yadav et al., 2023) | 20.48 | 84.40 | 81.13 | 58.19 | 85.70 | 64.65 | 74.81 | 42.96 | 64.04 | 25.15 | 0 |
| DARE (Yu et al., 2024) | 9.28 | 77.87 | 77.94 | 30.77 | 79.25 | 39.35 | 71.48 | 62.09 | 56.00 | 34.54 | 0 |
| SVD | 58.31 | 93.92 | 88.48 | 90.65 | 87.56 | 85.80 | 92.26 | 65.25 | 82.78 | 3.24 | 4.14 |
| Twin-Merging (Lu et al., 2024) | 59.12 | 93.53 | 88.65 | 72.36 | 89.17 | 84.30 | 92.32 | 73.89 | 81.67 | 4.55 | 100.0 |
| EMR-Merging (Huang et al., 2024) | 39.96 | 93.35 | 86.27 | 82.73 | 89.72 | 85.45 | 89.57 | 74.37 | 80.18 | 6.29 | 15.62 |
| TALL-Mask (Wang et al., 2024) | 45.81 | 93.81 | 88.73 | 88.87 | 88.51 | 80.29 | 92.37 | 75.09 | 81.69 | 4.52 | 15.62 |
| T-Switch (Qi et al., 2025) | 53.12 | 94.04 | 89.22 | 90.15 | 91.16 | 87.08 | 92.57 | 77.26 | 84.33 | 1.44 | 6.25 |
| DTS-T | 59.33 | 93.66 | 89.68 | 90.59 | 90.55 | 86.50 | 91.80 | 77.26 | 84.93 | 0.73 | 3.81 |
| DTS-D | 59.66 | 93.69 | 89.71 | 90.62 | 90.80 | 87.09 | 92.11 | 76.17 | 84.98 | 0.67 | 3.81 |
| DTS-T* | 59.71 | 93.35 | 89.95 | 90.54 | 88.87 | 85.33 | 91.69 | 76.90 | 84.54 | 1.18 | 0.88 |
| DTS-D* | 59.26 | 93.81 | 89.96 | 90.58 | 89.63 | 86.45 | 92.18 | 76.17 | 84.75 | 0.94 | 0.88 |

Table 3: Multi-task performance (%) when merging Qwen-14B models on three tasks. ADR refers to the Accuracy Drop Rate, and AMR denotes the Additional Memory Rate. The best result is highlighted in **bold**, and the second-best result is underlined.

| Method | MMLU | TruthfulQA | BBQ | Avg. ↑ | ADR ↓ | AMR ↓ |
|---|---|---|---|---|---|---|
| Individual | 68.36 | 54.35 | 93.53 | 72.08 | – | – |
| Weight-Averaging (Wortsman et al., 2022) | 67.11 | 50.02 | 82.32 | 66.48 | 7.77 | 0 |
| DARE (Yu et al., 2024) | 67.23 | 51.31 | 83.74 | 67.43 | 6.45 | 0 |
| Task-Arithmetic (Ilharco et al., 2023) | 66.63 | 53.38 | 78.24 | 66.08 | 8.32 | 0 |
| Ties-Merging (Yadav et al., 2023) | 67.28 | 50.02 | 84.10 | 67.13 | 6.86 | 0 |
| SVD | 67.99 | 52.45 | 91.71 | 70.72 | 1.89 | 5.39 |
| Twin-Merging (Lu et al., 2024) | 68.07 | 52.38 | 90.73 | 70.39 | 2.34 | 100.0 |
| EMR-Merging (Huang et al., 2024) | 67.94 | 52.50 | 91.02 | 70.49 | 2.21 | 15.62 |
| T-Switch (Qi et al., 2025) | 68.05 | 53.72 | 92.50 | 71.42 | 0.91 | 6.25 |
| DTS-T | 68.30 | 54.12 | 92.97 | 71.80 | 0.39 | 3.57 |
| DTS-D | 68.32 | 54.11 | 92.99 | 71.81 | 0.37 | 3.57 |
| DTS-T* | 68.20 | 53.99 | 92.90 | 71.70 | 0.53 | 0.92 |
| DTS-D* | 68.18 | 53.99 | 92.91 | 71.70 | 0.53 | 0.92 |

2016), and RTE (Giampiccolo et al., 2007). For natural language generation, we use Qwen-14B (Bai et al., 2023) as the backbone, evaluating on MMLU (Hendrycks et al., 2020), TruthfulQA (Lin et al., 2022), and BBQ (Parrish et al., 2022), in line with prior work (Lu et al., 2024). More implementation details are provided in Appendix C.1.

**Metrics**. We report both absolute performance (accuracy) and relative performance, measured by the *accuracy drop rate* (**ADR**), defined as the ratio of the accuracy gap between the merged and individually fine-tuned models to the accuracy of the individually fine-tuned model (serving as the upper bound). A lower drop rate indicates reduced performance degradation. To evaluate memory efficiency, we additionally report the *additional memory rate* (**AMR**), which quantifies the extra memory required to store task-specific information per task, beyond the storage of the merged model itself. Lower values correspond to more memory-efficient approaches.

**Experimental results**. Individual models require storing a fully fine-tuned model per task, and we omit their additional memory usage in comparisons. Table 1 and Table 7 present detailed comparisons of model performance and additional memory overhead for visual classification tasks. The following key observations can be made: (1) Basic model merging methods perform significantly worse than individual models. (2) Recent personalized merging approaches improve per-task performance by incorporating task-specific parameters. However, these methods often incur substantial memory costs. For instance, WEMOE (Tang et al., 2024) requires an extra 58.80% of the model size per task. In contrast, our method achieves comparable or superior performance with only $\sim 1\%$ additional memory per task. (3) Our method offers flexibility in balancing performance and memory usage through a single tunable sparsity coefficient, allowing it to adapt to varying deployment constraints. In comparison, methods such as EMR-Merging (Huang et al., 2024) and T-Switch (Qi et al., 2025) rely on fixed storage budgets and lack adaptability, limiting practical applicability in real-world scenarios.

For natural language processing tasks, as shown in Table 2 and Table 8, the results on RoBERTa and GPT-2 follow trends similar to those observed in visual classification tasks. Notably, our method achieves 99.06% of the individual model's performance on RoBERTa, requiring only 0.88% additional memory per task, demonstrating a favorable trade-off between efficiency and effectiveness. For

Table 4: Generalization results (%) on two unseen tasks when merging ViT-B/32 models on six tasks. The best result is highlighted in **bold**, and the second-best result is underlined.

| Method | Seen Tasks | | | | | | | Unseen Tasks | | |
| --- | --- | --- | --- | --- | --- | --- | --- | --- | --- | --- |
| | SUN397 | Cars | EuroSAT | GTSRB | MNIST | DTD | Avg. | RESISC45 | SVHN | Avg. |
| Individual | 74.49 | 77.73 | 99.80 | 98.73 | 99.69 | 79.36 | 88.30 | 98.22 | 97.46 | 97.84 |
| Fisher-Merging (Matena & Raffel, 2022) | 68.19 | 67.41 | 86.47 | 67.23 | 81.64 | 58.69 | 71.61 | 60.25 | 42.51 | 51.38 |
| RegMean (Jin et al., 2023) | 69.45 | 70.53 | 97.06 | 86.99 | 98.35 | 67.12 | 81.58 | 50.22 | 51.50 | 50.86 |
| Task-Arithmetic (Ilharco et al., 2023) | 65.28 | 63.68 | 87.17 | 76.18 | 94.24 | 56.47 | 73.84 | 52.43 | 45.27 | 48.85 |
| Ties-Merging (Yadav et al., 2023) | 68.27 | 65.93 | 81.22 | 70.01 | 89.07 | 56.02 | 71.75 | 60.36 | 47.34 | 53.85 |
| DARE (Yu et al., 2024) | 69.99 | 69.32 | 72.16 | 55.39 | 84.52 | 56.81 | 68.03 | 51.60 | 49.36 | 50.48 |
| AdaMerging (Yang et al., 2024) | 69.84 | 72.45 | 95.18 | 95.53 | 98.16 | 70.71 | 83.65 | 48.75 | **60.72** | 54.74 |
| SVD | 71.18 | 71.60 | 99.64 | 98.02 | 99.66 | 77.58 | 86.28 | 60.60 | 23.50 | 42.05 |
| Twin-Merging (Lu et al., 2024) | 71.76 | 69.20 | 73.24 | 93.37 | 99.64 | 72.25 | 79.91 | 52.43 | 45.27 | 48.85 |
| WEMOE (Tang et al., 2024) | **74.32** | **78.16** | 98.71 | **98.64** | 99.57 | 75.13 | 87.42 | 47.39 | 51.37 | 49.38 |
| EMR-Merging (Huang et al., 2024) | 71.81 | 74.61 | 99.32 | 98.40 | 99.63 | 75.85 | 86.60 | 28.95 | 49.80 | 39.38 |
| TALL-Mask (Wang et al., 2024) | 73.02 | 77.38 | 99.38 | 98.46 | 99.66 | 77.61 | 87.58 | 52.43 | 45.27 | 48.85 |
| T-Switch (Qi et al., 2025) | 74.05 | 77.32 | 99.52 | 98.41 | 99.56 | 78.55 | 87.90 | 60.60 | 23.50 | 42.05 |
| DTS-T | 74.15 | 76.85 | 99.66 | 98.34 | 99.63 | **79.03** | 87.94 | 61.35 | 49.11 | 55.23 |
| DTS-D | 74.15 | 76.87 | **99.78** | 98.63 | **99.67** | 78.78 | **87.98** | 60.91 | 49.91 | 55.41 |
| DTS-T* | 74.14 | 76.26 | 99.70 | 98.25 | 99.61 | 78.74 | 87.78 | 61.32 | 48.92 | 55.12 |
| DTS-D* | 74.07 | 76.62 | 99.76 | 98.58 | **99.67** | 78.83 | 87.92 | **61.42** | 49.57 | **55.50** |

Table 5: Generalization results (%) on three unseen tasks when merging GPT-2 models on four tasks. The best result is highlighted in **bold**, and the second-best result is underlined.

| Method | Seen Tasks | | | | | Unseen Tasks | | | |
| --- | --- | --- | --- | --- | --- | --- | --- | --- | --- |
| | CoLA | MNLI | MRPC | QNLI | Avg. | QQP | RTE | SST-2 | Avg. |
| Individual | 76.80 | 81.99 | 80.39 | 88.27 | 81.86 | 89.64 | 65.34 | 91.17 | 82.05 |
| Task-Arithmetic (Ilharco et al., 2023) | 68.26 | 68.44 | 72.54 | 60.69 | 67.48 | 67.12 | 43.68 | 50.71 | 53.84 |
| Ties-Merging (Yadav et al., 2023) | 63.08 | 79.93 | 34.06 | 71.16 | 62.06 | 70.36 | 57.03 | 51.26 | 59.55 |
| DARE (Yu et al., 2024) | 63.37 | 65.51 | 69.11 | 57.00 | 63.75 | 68.07 | 42.96 | 51.56 | 54.20 |
| SVD | 76.22 | 81.26 | 79.90 | 87.84 | 81.31 | 63.16 | 52.70 | 50.91 | 55.59 |
| Twin-Merging (Lu et al., 2024) | 76.27 | 80.03 | 79.65 | 87.53 | 80.87 | 69.73 | 55.29 | 51.14 | 58.72 |
| EMR-MERGING (Huang et al., 2024) | 73.63 | 81.74 | 80.14 | 87.04 | 80.64 | 69.94 | 54.87 | 51.12 | 58.64 |
| TALL-Mask (Wang et al., 2024) | 74.78 | 78.59 | 78.43 | 88.15 | 79.98 | 67.12 | 43.68 | 50.71 | 53.84 |
| WEMOE (Tang et al., 2024) | 72.30 | 80.24 | 77.59 | 83.06 | 78.30 | 69.16 | 56.93 | 51.02 | 58.37 |
| T-Switch (Qi et al., 2025) | 76.27 | **81.87** | 79.90 | 88.57 | 81.65 | 63.16 | 52.70 | 50.91 | 55.59 |
| DTS-T | **76.98** | 81.72 | 79.94 | 87.99 | 81.66 | 70.00 | 56.73 | 50.45 | 59.06 |
| DTS-D | 76.69 | 81.58 | 80.33 | **88.64** | **81.81** | 70.03 | **57.65** | 51.03 | 59.57 |
| DTS-T* | 76.72 | 81.84 | 79.94 | 88.15 | 81.66 | 70.11 | 56.49 | 51.45 | 59.35 |
| DTS-D* | 76.51 | 81.62 | 80.26 | 88.41 | 81.70 | **71.12** | 56.93 | **52.08** | **60.04** |

natural language generation tasks, as shown in Table 3, the results align with those observed in the visual classification and natural language processing tasks. These results confirm that the task-specific information extracted by DTS is both compact and crucial, effectively capturing the key information of each task while requiring minimal additional storage.

## 4.2 GENERALIZATION ON UNSEEN TASKS

**Baselines and datasets.** To evaluate the generalization ability of our method, we conduct experiments on unseen tasks using the same set of baselines as in previous sections, with a few differences. The details can be found in Appendix B.2. The dataset setup remains consistent with previous experiments, with minor modifications to assess generalization. For visual classification tasks, we designate RESISC45 (Cheng et al., 2017) and SVHN (Netzer et al., 2011) from the original set of eight datasets as unseen tasks, following the protocol in (Tang et al., 2024). This means their fine-tuned models are not accessed during the merging process. The remaining six datasets are treated as seen tasks, and their corresponding fine-tuned models are used in merging. For natural language processing tasks, we consider QQP (Chen et al., 2018), RTE (Giampiccolo et al., 2007), and SST-2 (Socher et al., 2013) as unseen tasks, while the remaining four tasks are treated as seen tasks. For natural language generation tasks, we treat BBQ as the unseen task and the remaining datasets as seen tasks.

**Experimental results**. The results for the ViT-B/32 and ViT-L/14 backbones are presented in Table 4 and Table 9, while those for GPT-2 and Qwen-14B backbones can be found in Table 5 and Table 10. Our method consistently outperforms all baselines on both seen and unseen tasks. Notably, while personalized merging methods generally perform better than basic approaches on seen tasks, they do not necessarily guarantee better generalization. For example, on the ViT-B/32 backbone, all personalized baselines underperform Ties-Merging (Yadav et al., 2023) on unseen tasks. This highlights a key limitation: personalized methods tend to optimize for task-specific performance on seen tasks, which may limit their transferability to unseen tasks. In contrast, our method demonstrates strong performance on both seen and unseen tasks.

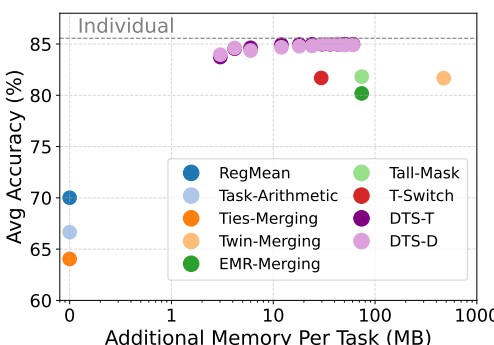

Figure 4: Performance (%) of merged model and additional memory requirement for different methods with RoBERTa as the backbone.

| Method | QQP | RTE | SST-2 | Avg. |
|---|---|---|---|---|
| EMR-Merging | 71.48 | 52.70 | 50.80 | 58.33 |
| WEMOE | **72.33** | 53.01 | 50.72 | 58.69 |
| Twin-Merging | 71.84 | 56.05 | 50.27 | 59.38 |
| T-Switch | 72.15 | 53.42 | 50.91 | 58.83 |
| DTS-T | 70.00 | 56.73 | 50.45 | 59.06 |
| DTS-D | 70.03 | **57.65** | 51.03 | 59.57 |
| DTS-T* | 70.11 | 56.49 | 51.45 | 59.35 |
| DTS-D* | 71.12 | 56.93 | **52.08** | **60.04** |

Table 6: Effectiveness of the task-specific information extracted by DTS. Even with the same weighting mechanism, our method consistently achieves the best performance on unseen tasks. The best result is highlighted in **bold**, and the second-best result is underlined.

### 4.3 ANALYSIS OF DTS

**Sensitivity to sparsity factor $r$.** In our approach, we approximate the parameters of the individual model by retaining only a partial rank of the SVD decomposition, controlled by the sparsity coefficient $r$. As shown in Fig. 4, we plot the relationship between performance and model storage overhead for different values of $r$, and compare our method against several baselines. Additionally, we present the performance of our method for varying values of $r$ in Table 13. Our method consistently achieves superior performance with minimal additional storage across all settings. Notably, in practical deployments, setting $r$ within the range $[0, 0.5]$ is sufficient to achieve strong performance while maintaining low memory overhead.

**Effectiveness of the task-specific information extracted by DTS.** To further validate the effectiveness of the task-specific information extracted by our method, we compare it with several existing personalized merging approaches, integrated with our adaptive weighting strategy on unseen tasks. Specifically, for EMR-Merging (Huang et al., 2024), we use the variant of DTS to combine the reconstructed personalized parameters, resulting in a model adapted for unseen tasks. For WEMOE (Tang et al., 2024) and T-Switch (Qi et al., 2025), we apply our method to the personalized MLP layers and binarized parameters obtained from seen tasks, respectively. Using the same experimental setup as in the GPT-2 experiments, we report results on unseen tasks in Table 6. Under the same adaptive merging mechanism, our method consistently achieves the best performance, further demonstrating the effectiveness and generalization of the task-specific information captured by DTS.

**Effectiveness of each component in DTS.** Our method is composed of three key components: decomposition, thresholding, and scaling. As shown in Table 11, without decomposition, storing task-specific information would require significant extra memory, although it would not affect performance. To demonstrate the effectiveness of thresholding and scaling, we conducted an ablation study, with results presented in Table 12. Compared to simple binarization, our thresholding approach yields a performance improvement of approximately 2%. Additionally, without the scaling strategy, model performance drops to around 5%. These experimental results underscore the efficiency and effectiveness of each component in our method.

## 5 CONCLUSION

In this work, we revisited the challenge of model merging from the perspective of task similarity and demonstrated that significant performance degradation persists even when merging models fine-tuned on highly similar tasks. To address this, we introduced DTS—a compact and effective model merging method based on decomposition, thresholding, and scaling. DTS efficiently preserves essential task-specific information by decomposing task vectors into low-rank approximations, achieving high performance with minimal memory overhead. To support generalization to unseen tasks, we further proposed a variant of DTS, a data-free adaptive merging strategy that weights task-specific information based on the semantic similarity of task characteristics. Experimental results on standard multi-task model merging benchmarks demonstrate that our method consistently outperforms state-of-the-art baselines, requiring only 1% extra storage per task. Moreover, experiments on unseen tasks show that the adaptive variant of DTS achieves superior generalization performance.

## 6 ETHICS STATEMENT

This study adheres to the ICLR Code of Ethics, and we confirm that there are no ethical concerns associated with this research. No human subjects or animal experimentation were involved in the study. All datasets used were sourced in compliance with relevant guidelines, ensuring no violation of privacy or security. The research does not include any personally identifiable information, and no experiments were conducted that could raise privacy, security, or legal concerns. As such, we affirm that the work presented in this paper adheres to the ethical standards and complies with all relevant ethical guidelines.

## 7 REPRODUCIBILITY STATEMENT

To ensure the reproducibility of our work, we have made all necessary resources and information readily available. A comprehensive description of our method, DTS, is provided in Appendix A, where we detail the step-by-step procedure. Additionally, the source code for implementing our method is included as supplementary material to facilitate the easy reproduction of our results. The experimental setup is described in detail in the paper.

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

# Technical Appendices and Supplementary Material

## A   THE COMPLETE PIPELINE OF DTS

Here, we take the task vector as an example to illustrate the complete pipeline of DTS, which can also be applied to the difference vector.

We first employ singular value decomposition on each task vector and retain only the top-r singular values as follows:

$$\boldsymbol{U}_n, \boldsymbol{\Sigma}_n, \boldsymbol{V}_n = \text{SVD}_r(\boldsymbol{\tau}_n), \tag{9}$$

where $\boldsymbol{U}_n$ and $\boldsymbol{V}_n$ are the left and right singular vector matrices, and $\boldsymbol{\Sigma}_n$ contains the singular values. Here, r is the proportion of singular values retained.

To further reduce storage, we threshold the decomposed components. Specifically, take $\boldsymbol{U}_n$ as an example, we mark the positive and negative distribution of all parameters as follows:

$$g(U_{n,j}) = \begin{cases} 1, & \text{if } U_{n,j} > 0, \\ 0, & \text{otherwise}, \end{cases} \tag{10}$$

where $U_{n,j}$ is the $j$-th element of $\boldsymbol{U}_n$. Then, we divide the positive and negative values into two groups based on their magnitudes as follows:

$$g^+(U_{n,j}) = \begin{cases} 1, & \text{if } U_{n,j} > \lambda_1, \\ 0, & \text{otherwise}, \end{cases} \quad g^-(U_{n,j}) = \begin{cases} 1, & \text{if } \lambda_2 < U_{n,j} < 0, \\ 0, & \text{otherwise}, \end{cases} \tag{11}$$

where $\lambda_1$ and $\lambda_2$ are the medians of the positive and negative values in $\boldsymbol{U}_n$, respectively. Finally, a scaling factor is computed for each group:

$$s_1^+ = \frac{\|\boldsymbol{U}_n \odot g(\boldsymbol{U}_n) \odot g^+(\boldsymbol{U}_n)\|_2}{\|g(\boldsymbol{U}_n) \odot g^+(\boldsymbol{U}_n)\|_2}, \qquad s_2^+ = \frac{\|\boldsymbol{U}_n \odot g(\boldsymbol{U}_n) \odot (1 - g^+(\boldsymbol{U}_n))\|_2}{\|g(\boldsymbol{U}_n) \odot (1 - g^+(\boldsymbol{U}_n))\|_2},$$

$$s_1^- = \frac{\|\boldsymbol{U}_n \odot g(\boldsymbol{U}_n) \odot g^-(\boldsymbol{U}_n)\|_2}{\|g(\boldsymbol{U}_n) \odot g^-(\boldsymbol{U}_n)\|_2}, \qquad s_2^- = \frac{\|\boldsymbol{U}_n \odot g(\boldsymbol{U}_n) \odot (1 - g^-(\boldsymbol{U}_n))\|_2}{\|g(\boldsymbol{U}_n) \odot (1 - g^-(\boldsymbol{U}_n))\|_2}. \tag{12}$$

During inference, the reconstructed $\hat{\boldsymbol{U}}_n$ for each task is computed as follows:

$$\begin{aligned} \hat{\boldsymbol{U}}_n =& s_1^+ \cdot g(\boldsymbol{U}_n) \odot g^+(\boldsymbol{U}_n) + s_2^+ \cdot g(\boldsymbol{U}_n) \odot (1 - g^+(\boldsymbol{U}_n)) \\ &+ s_1^- \cdot g(\boldsymbol{U}_n) \odot g^-(\boldsymbol{U}_n) + s_2^- \cdot g(\boldsymbol{U}_n) \odot (1 - g^-(\boldsymbol{U}_n)) \end{aligned} \tag{13}$$

Similarly, during the approximation process, we apply the same thresholding strategy to $\boldsymbol{V}_n$, and obtain the approximated $\hat{\boldsymbol{V}}_n$ during inference. Since $\boldsymbol{\Sigma}_n$ only requires storing its diagonal values, which takes up minimal storage, we do not apply any approximation to $\boldsymbol{\Sigma}_n$.

Finally, during the inference stage, we reconstruct the approximated model parameters for the $n$-th task as follows:

$$\hat{\boldsymbol{\theta}}_n = \boldsymbol{\theta}_0 + \hat{\boldsymbol{U}}_n \times \boldsymbol{\Sigma}_n \times \hat{\boldsymbol{V}}_n. \tag{14}$$

It is worth noting that in our method, by leveraging singular value decomposition and thresholding-based grouped approximation, we only need to store six mask matrices and eight scaling factors for each task to reconstruct $\boldsymbol{U}_n$ and $\boldsymbol{V}_n$, along with the singular values. This significantly reduces the storage cost of task-specific information. Moreover, the experimental results demonstrate that our method outperforms all state-of-the-art approaches, needing only 1% extra storage per task.

## B   BASELINES

### B.1   BASELINES FOR SEEN TASKS

- **Individual Models** refer to task-specific models before merging.

- **Weight-Averaging** performs element-wise averaging of the weights across all fine-tuned models.

- **Fisher-Merging** (Matena & Raffel, 2022) leverages Fisher information matrices to estimate parameter importance and merges model weights using importance-weighted averaging.

- **RegMean** (Jin et al., 2023) merges models via a closed-form solution to a least-squares problem. Given $K$ linear models with weights $\boldsymbol{W}_i$ such that $f_i(\boldsymbol{x}) = \boldsymbol{W}_i^\top \boldsymbol{x}$, the objective is $\min_{\boldsymbol{W}} \sum_{i=1}^{K} \|\boldsymbol{W}^\top \boldsymbol{X}_i - \boldsymbol{W}_i^\top \boldsymbol{X}_i\|^2$, where $\boldsymbol{X}_i$ denotes the input features for the $i$-th model. The closed-form solution is:

$$\boldsymbol{W} = \left( \sum_{i=1}^{K} \boldsymbol{X}_i^\top \boldsymbol{X}_i \right)^{-1} \left( \sum_{i=1}^{K} \boldsymbol{X}_i^\top \boldsymbol{X}_i \boldsymbol{W}_i \right).$$

  The merging process requires computing input inner-product matrices $\boldsymbol{X}_i^\top \boldsymbol{X}_i$ in advance.

- **Task-Arithmetic** (Ilharco et al., 2023) defines task vectors as the difference between fine-tuned model weights and the pre-trained model weights, i.e., $\boldsymbol{\tau}_i = \boldsymbol{\theta}_i - \boldsymbol{\theta}_0$. To merge $K$ models $\{\boldsymbol{\theta}_i\}_{i=1}^{K}$, the merged model is computed as:

$$\boldsymbol{\theta}_M = \boldsymbol{\theta}_0 + \lambda \sum_{i=1}^{K} \boldsymbol{\tau}_i,$$

  where $\lambda$ is a tunable merging coefficient.

- **Ties-Merging** (Yadav et al., 2023) attributes performance degradation to conflicts among task vectors and addresses this issue by removing redundant parameters and resolving directional inconsistencies.

- **DARE** (Yu et al., 2024) leverages the redundancy in language models by randomly dropping a large portion (up to 90%–99%) of the delta parameters prior to merging, thereby reducing inter-model interference.

- **AdaMerging** (Yang et al., 2024) learns merging coefficients in an unsupervised manner, either at the task level (Task-wise AdaMerging) or at the layer level (Layer-wise AdaMerging). **AdaMerging++** further incorporates Ties-Merging (Yadav et al., 2023) as a preprocessing step before learning the coefficients.

- **SVD** applies singular value decomposition to each task vector matrix and retains only the top-$r$ singular values, while one-dimensional parameter vectors remain unchanged. To ensure a fair comparison, we control $r$ so that the additional storage required by SVD in the baseline is comparable to that of our method, and we present the performance under these conditions for SVD.

- **Twin-Merging** (Lu et al., 2024) stores complete model parameters for each task and employs a routing mechanism that performs weighted parameter fusion based on learned expert distributions.

- **WEMOE** (Tang et al., 2024) transforms MLP layers into a mixture-of-experts (MoE) structure via test-time adaptation, while merging the remaining parameters using Task-Arithmetic (Ilharco et al., 2023).

- **EMR-Merging** (Tang et al., 2024) selects a unified base model through Task-Arithmetic (Ilharco et al., 2023) and generates lightweight task-specific modulators—including binary masks and scaling factors—to align both the direction and magnitude of each personalized model with the unified base.

- **TALL-Mask** (Wang et al., 2024) uses a data-driven method to identify and eliminate task-irrelevant information from the merged multi-task vector. The resulting task-specific binary masks are used to localize and extract the knowledge relevant to each individual task.

- **T-Switch** (Qi et al., 2025) filters out parameters below a predefined threshold, binarizes the remaining ones based on sign, and applies separate scaling factors to positive and negative groups to approximate task-specific contributions.

### B.2 Baselines for unseen tasks

For basic merging methods, the merged model is directly used for inference on both seen and unseen tasks. For personalized merging methods, we follow the evaluation protocols outlined in the respective original papers for seen tasks. For unseen tasks, we adopt the following strategies based on each baseline's design: we use the merged model for Twin-Merging (Lu et al., 2024) and WEMOE (Tang et al., 2024), the unified model for EMR-Merging (Huang et al., 2024), and the original pretrained model for simple SVD and T-Switch (Qi et al., 2025).

## C More Experimental Details

### C.1 Datasets

For visual classification tasks, we employ classification accuracy as the evaluation metric. For natural language processing tasks, we follow the standard GLUE evaluation protocol: CoLA is assessed using Matthews correlation, STS-B using the average of Pearson and Spearman correlations, and all other tasks using classification accuracy. For natural language generation tasks, we use the same evaluation metrics as those in previous work (Lu et al., 2024).

## D More Experimental Results

### D.1 More Backbones

In addition to the backbones evaluated in the main paper, we also assess the performance of various methods on ViT-L/14 and GPT-2 backbones. As shown in Table 7 and Tables 8, our method consistently outperforms the baselines under the conventional multi-task model merging setting, demonstrating the effectiveness and robustness of our approach.

Moreover, for generalization on unseen tasks, we also evaluate the results using ViT-L/14 and Qwen-14B as backbones, as shown in Table 9 and Table 10. Our method consistently outperforms all baselines on both seen and unseen tasks.

Table 7: Multi-task performance (%) when merging ViT-L/14 models on eight tasks. ADR refers to the Accuracy Drop Rate, and AMR denotes the Additional Memory Rate. The best result is highlighted in **bold**, and the second-best result is underlined.

| Method | SUN397 | Cars | RESISC45 | EuroSAT | SVHN | GTSRB | MNIST | DTD | Avg. ↑ | ADR ↓ | AMR ↓ |
|---|---|---|---|---|---|---|---|---|---|---|---|
| Individual | 81.72 | 92.39 | 98.85 | 99.88 | 98.11 | 99.24 | 99.69 | 84.15 | 94.25 | – | – |
| Weight-Averaging (Wortsman et al., 2022) | 72.19 | 81.42 | 82.55 | 91.93 | 78.08 | 70.76 | 97.14 | 62.95 | 79.63 | 15.52 | 0 |
| Fisher-Merging (Matena & Raffel, 2022) | 69.24 | 88.61 | 87.50 | 93.53 | 80.66 | 74.82 | 93.32 | 70.07 | 82.22 | 12.77 | 0 |
| RegMean (Jin et al., 2023) | 73.38 | 81.80 | 86.10 | 97.01 | 88.12 | 84.27 | 98.54 | 60.82 | 83.76 | 11.14 | 0 |
| Task-Arithmetic (Ilharco et al., 2023) | 73.92 | 82.13 | 87.64 | 92.82 | 87.91 | 86.77 | 98.94 | 65.64 | 84.47 | 10.38 | 0 |
| Ties-Merging (Yadav et al., 2023) | 74.74 | 84.50 | 89.00 | 94.18 | 85.66 | 82.07 | 98.65 | 67.71 | 84.56 | 10.28 | 0 |
| DARE (Yu et al., 2024) | 73.03 | 82.70 | 86.19 | 93.41 | 85.26 | 83.48 | 98.58 | 65.69 | 83.54 | 11.36 | 0 |
| AdaMerging (Yang et al., 2024) | 79.03 | 90.34 | 90.86 | 96.19 | 93.44 | 98.05 | 99.12 | 79.94 | 90.87 | 3.59 | 0 |
| AdaMerging++ (Yang et al., 2024) | 79.46 | 90.38 | 91.66 | 97.47 | 93.42 | 97.55 | 99.05 | 79.20 | 91.02 | 3.43 | 0 |
| SVD | 78.49 | 89.73 | 98.02 | 99.86 | 98.08 | 98.91 | 99.71 | 81.91 | 93.09 | 1.23 | 4.08 |
| Twin-Merging (Lu et al., 2024) | **81.92** | 91.59 | 96.87 | 99.72 | 98.03 | 92.42 | 99.57 | 83.94 | 93.01 | 1.31 | 100.0 |
| WEMOE (Tang et al., 2024) | 81.42 | 92.10 | 95.46 | 99.48 | 97.73 | 99.13 | 99.70 | 83.74 | 93.60 | 0.70 | 58.80 |
| EMR-Merging (Huang et al., 2024) | 80.47 | 90.71 | 98.55 | 99.54 | 97.94 | 99.10 | 99.69 | 82.71 | 93.59 | 0.71 | 15.62 |
| TALL-Mask (Wang et al., 2024) | 80.58 | 91.61 | 98.68 | 99.76 | 98.08 | 99.24 | 99.74 | 83.14 | 93.85 | 0.42 | 15.62 |
| T-Switch (Qi et al., 2025) | 81.84 | **92.38** | **98.89** | 99.74 | 98.03 | 99.08 | 99.63 | 83.72 | 94.16 | 0.10 | 6.25 |
| DTS-T | 81.82 | 91.93 | 98.82 | **99.90** | 98.05 | 99.21 | **99.77** | 83.93 | 94.18 | 0.08 | 2.95 |
| DTS-D | 81.75 | 92.09 | **98.89** | 99.88 | 99.27 | **99.25** | 99.72 | **84.15** | **94.24** | **0.02** | 2.95 |
| DTS-T* | 81.72 | 92.17 | 98.83 | 99.82 | 98.03 | 99.15 | 99.71 | 83.87 | 94.16 | 0.10 | 0.99 |
| DTS-D* | 81.69 | 91.73 | 98.77 | 99.86 | 98.15 | 99.22 | 99.72 | 83.99 | 94.14 | 0.12 | 0.99 |

### D.2 Ablation on Decomposition, Thresholding, and Scaling Strategies

Our method primarily consists of decomposition, thresholding, and scaling strategies. Without decomposition, as shown in Table 11, storing task-specific information would require substantial additional memory, although it would not lead to a performance drop. In the thresholding strategy, we first apply sign-based thresholding to the model parameters, followed by separate thresholding of the positive and negative components, each further divided into two groups. This design is intended to preserve finer-grained task-specific information. To evaluate the effectiveness of this thresholding

Table 8: Multi-task performance (%) when merging GPT-2 models on seven tasks. ADR refers to the Accuracy Drop Rate, and AMR denotes the Additional Memory Rate. The best result is highlighted in **bold**, and the second-best result is underlined.

| Method | CoLA | MNLI | MRPC | QNLI | QQP | RTE | SST-2 | Avg. ↑ | ADR ↓ | AMR ↓ |
|---|---|---|---|---|---|---|---|---|---|---|
| Individual | 76.80 | 81.99 | 80.39 | 88.27 | 89.64 | 65.34 | 91.17 | 81.94 | – | – |
| Weight-Averaging (Wortsman et al., 2022) | 55.03 | 55.16 | 51.21 | 57.65 | 76.71 | 44.76 | 52.53 | 56.15 | 31.48 | 0 |
| Fisher-Merging (Matena & Raffel, 2022) | 54.81 | 58.12 | 39.53 | 63.28 | 81.46 | 49.12 | 64.74 | 58.72 | 28.34 | 0 |
| RegMean (Jin et al., 2023) | 61.69 | 70.44 | 65.37 | 69.71 | 78.83 | 56.10 | 79.74 | 68.84 | 15.99 | 0 |
| Task-Arithmetic (Ilharco et al., 2023) | 68.71 | 68.64 | 69.56 | 70.42 | 81.83 | 47.21 | 83.62 | 70.00 | 14.58 | 0 |
| Ties-Merging (Yadav et al., 2023) | 68.48 | 71.46 | 68.49 | 69.56 | 82.57 | 47.68 | 81.82 | 70.01 | 14.56 | 0 |
| DARE (Yu et al., 2024) | 67.59 | 65.41 | 72.54 | 62.07 | 79.74 | 44.76 | 72.46 | 66.37 | 19.01 | 0 |
| SVD | 74.49 | 80.91 | 79.41 | 88.12 | 88.21 | 64.14 | 91.16 | 80.92 | 1.24 | 3.80 |
| Twin-Merging (Lu et al., 2024) | 76.02 | 78.75 | 78.93 | 87.17 | 87.58 | 62.82 | 90.16 | 80.20 | 2.12 | 100.0 |
| EMR-Merging (Huang et al., 2024) | 72.77 | 81.08 | 79.16 | 84.84 | 88.11 | 66.43 | 90.25 | 80.38 | 1.91 | 17.41 |
| TALL-Mask (Wang et al., 2024) | 74.78 | 78.59 | 78.43 | 85.61 | 88.91 | **67.50** | 90.82 | 80.55 | 1.82 | 17.41 |
| T-Switch (Qi et al., 2025) | 76.27 | **81.87** | 79.90 | **88.57** | 88.54 | 63.17 | 90.82 | 81.31 | 0.78 | 6.23 |
| DTS-T | **76.98** | 81.72 | 79.94 | 87.99 | 89.20 | 65.34 | **91.16** | 81.76 | 0.22 | 5.24 |
| DTS-D | 76.69 | 81.58 | **80.63** | 88.24 | **89.46** | 64.98 | 91.05 | **81.80** | 0.17 | 5.24 |
| DTS-T* | 76.60 | 81.86 | 79.94 | 87.95 | 88.18 | 64.62 | 90.13 | 81.33 | 0.75 | 0.93 |
| DTS-D* | 76.40 | 81.68 | 80.41 | 88.44 | 88.81 | 66.06 | 90.94 | 81.39 | 0.67 | 0.93 |

Table 9: Generalization results (%) on two unseen tasks when merging ViT-L/14 models on six tasks. The best result is highlighted in **bold**, and the second-best result is underlined.

| | Seen Tasks | | | | | | | Unseen Tasks | | |
|---|---|---|---|---|---|---|---|---|---|---|
| Method | SUN397 | Cars | EuroSAT | GTSRB | MNIST | DTD | Avg | RESISC45 | SVHN | Avg. |
| Individual | 81.72 | 92.39 | 99.88 | 99.24 | 99.69 | 84.15 | 92.85 | 98.85 | 98.11 | 98.48 |
| Fisher-Merging (Matena & Raffel, 2022) | 68.92 | 88.61 | 94.93 | 82.67 | 90.05 | 72.41 | 82.93 | 70.55 | 64.75 | 67.65 |
| RegMean (Jin et al., 2023) | 77.83 | 89.92 | 98.53 | 92.62 | 98.90 | 78.71 | 89.42 | 59.94 | 75.52 | 67.73 |
| Task-Arithmetic (Ilharco et al., 2023) | 75.28 | 85.70 | 95.50 | 89.37 | 98.91 | 69.20 | 85.66 | 69.02 | 69.89 | 69.46 |
| Ties-Merging (Yadav et al., 2023) | 77.21 | 88.37 | 96.74 | 92.44 | 99.33 | 74.63 | 88.12 | 66.00 | 72.32 | 69.16 |
| DARE (Yu et al., 2024) | 76.56 | 84.45 | 97.00 | 90.46 | 99.58 | 71.12 | 86.53 | 71.87 | 65.80 | 68.84 |
| AdaMerging (Yang et al., 2024) | 79.98 | 90.34 | 97.14 | 98.47 | 99.30 | 80.52 | 90.96 | 64.18 | **78.57** | 71.38 |
| SVD | 78.49 | 89.73 | 99.86 | 98.91 | 99.71 | 81.91 | 91.43 | 71.33 | 58.45 | 64.89 |
| Twin-Merging (Lu et al., 2024) | **81.95** | 91.60 | 99.73 | 93.42 | 99.59 | 83.54 | 91.64 | 69.02 | 69.89 | 69.46 |
| WEMOE (Tang et al., 2024) | 81.64 | 92.22 | 99.53 | 99.14 | 99.68 | 82.85 | 92.51 | 61.36 | 76.58 | 68.97 |
| EMR-Merging (Huang et al., 2024) | 80.87 | 91.52 | 99.64 | 99.10 | 99.73 | 83.24 | 92.35 | 60.03 | 70.60 | 65.32 |
| TALL-Mask (Wang et al., 2024) | 80.58 | 91.61 | 99.76 | 99.24 | 99.74 | 83.14 | 92.34 | 69.02 | 69.89 | 69.46 |
| T-Switch (Qi et al., 2025) | 81.84 | **92.38** | 99.74 | 99.08 | 99.63 | 83.72 | 92.73 | 71.33 | 58.45 | 64.89 |
| DTS-T | 81.82 | 91.93 | **99.90** | 99.21 | **99.77** | 83.93 | 92.76 | 72.90 | 71.80 | 72.35 |
| DTS-D | 81.75 | 92.09 | 99.88 | **99.25** | 99.72 | **84.15** | 92.81 | 72.58 | 72.01 | 72.30 |
| DTS-T* | 81.72 | 92.24 | 99.78 | 99.12 | 99.67 | 83.85 | 92.73 | **72.98** | 71.95 | 72.47 |
| DTS-D* | 81.79 | 92.07 | 99.86 | 99.22 | 99.72 | 83.99 | 92.78 | 72.75 | 72.69 | **72.72** |

strategy, we conduct an ablation study comparing our proposed method with a baseline of binarization using ViT-B/32 as the backbone. The binarization strategy divides model parameters into positive and negative components, assigning a single scaling factor to each. As shown in Table 12, our fine-grained thresholding significantly outperforms the single-step approach, demonstrating its advantage in retaining valuable task-specific information. Additionally, to demonstrate the effectiveness of our scaling strategy, we conduct an ablation study, as shown in Table 12. Without the scaling strategy, the model performance drops significantly, highlighting the importance of our approach.

### D.3 SENSITIVITY ANALYSIS ON SPARSE FACTOR $r$

As shown in Table 13, we report the accuracy of DTS-D under varying sparsity factors $r$ on the ViT-B/32 and RoBERTa backbones. Notably, when $r < 0.5$, increasing the number of preserved parameters consistently improves performance. However, for $r > 0.5$, the gains begin to plateau, indicating diminishing returns with additional parameter retention. These results suggest that, for practical deployments, setting $r$ within the range $[0, 0.5]$ is sufficient to achieve strong performance while maintaining low memory overhead. Finally, in practical deployments, setting $r$ within the range $[0, 0.5]$ is sufficient to achieve strong performance while maintaining low memory overhead.

### D.4 MERGING MODELS WITH THE SAME TASKS

To further support the observation made in the main paper—that even for similar tasks, merged models often exhibit substantial performance gaps compared to individually fine-tuned models—we

Table 10: Generalization results (%) on unseen tasks BBQ when merging Qwen-14B models on two seen tasks. The best result is highlighted in **bold**, and the second-best result is underlined.

| | Seen Tasks | | | Unseen Task |
|---|---|---|---|---|
| Method | MMLU | TruthfulQA | Avg. | BBQ |
| Individual | 68.36 | 54.35 | 61.36 | 94.53 |
| DARE (Yu et al., 2024) | 67.82 | 52.79 | 60.31 | 85.96 |
| Task-Arithmetic (Ilharco et al., 2023) | 67.05 | 53.52 | 60.29 | 84.88 |
| Ties-Merging (Yadav et al., 2023) | 67.74 | 51.46 | 59.60 | 85.01 |
| Twin-Merging (Lu et al., 2024) | 68.14 | 52.78 | 60.46 | 86.46 |
| EMR-MERGING (Huang et al., 2024) | 68.00 | 52.91 | 60.46 | 86.11 |
| T-Switch (Qi et al., 2025) | 68.05 | 53.72 | 60.89 | 80.69 |
| DTS-T | 68.30 | **54.12** | 61.21 | 87.12 |
| DTS-D | **68.32** | 54.11 | **61.22** | 87.09 |
| DTS-T* | 68.20 | 53.99 | 61.10 | 87.28 |
| DTS-D* | 68.10 | 53.99 | 61.05 | **87.30** |

Table 11: Ablation results (%) on decomposition strategies with ViT-B/32 as the backbone. AMR denotes the Additional Memory Rate.

| Method | SUN397 | Cars | RESISC45 | EuroSAT | SVHN | GTSRB | MNIST | DTD | Avg. ↑ | AMR ↓ |
|---|---|---|---|---|---|---|---|---|---|---|
| DTS-T | 74.15 | 76.85 | 97.92 | 99.66 | 97.00 | 98.34 | 99.63 | 79.03 | 90.32 | 3.68 |
| Without Decomposition | 74.39 | 76.51 | 98.15 | 99.80 | 97.33 | 98.51 | 99.61 | 78.94 | 90.41 | 9.37 |
| DTS-D | 74.15 | 76.87 | 97.98 | 99.78 | 97.36 | 98.63 | 99.67 | 78.78 | 90.40 | 3.68 |
| Without Decomposition | 74.11 | 76.82 | 98.16 | 99.78 | 97.44 | 98.57 | 99.67 | 78.94 | 90.43 | 9.37 |

Table 12: Ablation results (%) on thresholding and scaling strategies with ViT-B/32 as the backbone. Without thresholding, we employ binarization as an alternative.

| Method | SUN397 | Cars | RESISC45 | EuroSAT | SVHN | GTSRB | MNIST | DTD | Avg. |
|---|---|---|---|---|---|---|---|---|---|
| DTS-T | 74.15 | 76.85 | 97.92 | 99.66 | 97.00 | 98.34 | 99.63 | 79.03 | 90.32 |
| Without Thresholding | 71.85 | 71.98 | 97.21 | 99.50 | 95.77 | 97.83 | 99.48 | 74.51 | 88.52 |
| Without Scaling | 0.43 | 0.58 | 3.19 | 10.64 | 6.95 | 3.50 | 10.92 | 3.65 | 4.98 |
| DTS-D | 74.15 | 76.87 | 97.98 | 99.78 | 97.36 | 98.63 | 99.67 | 78.78 | 90.40 |
| Without Thresholding | 72.53 | 72.00 | 97.56 | 99.68 | 95.98 | 97.95 | 99.59 | 74.71 | 88.75 |
| Without Scaling | 0.44 | 0.81 | 3.88 | 11.65 | 10.71 | 5.68 | 10.70 | 3.05 | 5.87 |

Table 13: Sensitivity Analysis (%) on the sparse coefficient $r$. The table reports the average accuracy on the benchmark datasets for the ViT-B/32 and RoBERTa backbones, respectively.

| $r$ | 0.05 | 0.07 | 0.1 | 0.2 | 0.3 | 0.4 | 0.5 | 0.6 | 0.7 | 0.8 | 0.9 | 1.0 |
|---|---|---|---|---|---|---|---|---|---|---|---|---|
| ViT/B-32 | 89.68 | 90.08 | 90.18 | 90.25 | 90.32 | 90.37 | 90.41 | 90.40 | 90.39 | 90.40 | 90.40 | 90.41 |
| RoBERTa | 83.73 | 84.49 | 84.58 | 84.63 | 85.93 | 84.94 | 84.97 | 84.97 | 84.98 | 84.96 | 90.4 | 84.98 |

provide additional analysis below. As shown in Table 14, the performance of the merged model is significantly lower than that of the fine-tuned model on both digit classification and single-sentence tasks. This underscores the importance of preserving task-specific information during the model merging process.

Table 14: Merging results (%) on different datasets with the same task under various backbones.

| | Digit Classification (ViT/B-32) | | | Digit Classification (ViT/L-14) | | | Similarity & Paraphrase Tasks (RoBERTa) | | | |
|---|---|---|---|---|---|---|---|---|---|---|
| Method | MNIST | SVHN | Avg. | MNIST | SVHN | Avg. | MRPC | STSB | QQP | Avg. |
| Individual | 99.69 | 97.46 | 98.58 | 99.69 | 98.11 | 98.90 | 89.22 | 90.63 | 91.41 | 90.42 |
| DARE (Yu et al., 2024) | 96.15 | 88.68 | 92.42 | 99.48 | 94.62 | 97.05 | 81.57 | 55.21 | 81.94 | 72.91 |
| Task-Arithmetic (Ilharco et al., 2023) | 99.41 | 92.85 | 96.13 | 95.80 | 97.66 | 96.13 | 83.82 | 68.08 | 85.53 | 79.14 |
| Ties-Merging (Yadav et al., 2023) | 99.29 | 93.14 | 96.22 | 99.45 | 94.51 | 96.98 | 82.84 | 61.75 | 87.49 | 77.36 |
| DTS-T | 99.70 | 97.44 | 98.57 | 99.77 | 98.06 | 98.91 | 89.96 | 90.60 | 91.14 | 90.56 |
| DTS-D | 99.69 | 97.46 | 98.57 | 99.73 | 99.27 | 99.50 | 89.72 | 90.63 | 90.91 | 90.42 |

Table 15: Comparison of performance (%) between task vector (-task) and difference vector (-diff) applications to baseline methods for merging ViT-B/32 models on eight tasks. (The values) represent the performance gains.

| Method | SUN397 | Cars | RESISC45 | EuroSAT | SVHN | GTSRB | MNIST | DTD | Avg. |
|---|---|---|---|---|---|---|---|---|---|
| Individual | 74.49 | 77.73 | 98.22 | 99.80 | 97.46 | 98.73 | 99.69 | 79.36 | 90.69 |
| Weight Averaging (Wortsman et al., 2022) | 65.35 | 63.41 | 71.42 | 71.69 | 64.20 | 52.82 | 87.56 | 50.18 | 65.83 |
| Task-Arithmetic (Ilharco et al., 2023)-task | 54.78 | 54.98 | 67.69 | 78.70 | 80.21 | 69.68 | 97.34 | 50.37 | 69.22 |
| Task-Arithmetic (Ilharco et al., 2023)-diff | 64.65 | 63.26 | 72.10 | 71.88 | 64.16 | 52.80 | 87.46 | 50.74 | 65.88 (-3.34) |
| Ties-Merging (Yadav et al., 2023)-task | 64.17 | 64.43 | 76.31 | 76.62 | 81.28 | 69.37 | 96.53 | 54.52 | 72.90 |
| Ties-Merging (Yadav et al., 2023)-diff | 65.65 | 63.56 | 71.82 | 68.44 | 62.83 | 51.62 | 87.90 | 51.60 | 65.43 (-7.47) |
| EMR-Merging (Huang et al., 2024)-task | 71.02 | 72.75 | 93.49 | 99.24 | 96.86 | 98.12 | 99.58 | 74.36 | 88.18 |
| EMR-Merging (Huang et al., 2024)-diff | 74.73 | 77.98 | 98.00 | 99.72 | 97.27 | 98.73 | 99.66 | 79.15 | 90.66 (+2.48) |
| T-Switch (Qi et al., 2025)-task | 74.05 | 77.32 | 96.47 | 99.52 | 97.33 | 98.41 | 99.56 | 78.55 | 90.15 |
| T-Switch (Qi et al., 2025)-diff | 74.72 | 77.85 | 98.13 | 99.70 | 97.39 | 98.85 | 99.69 | 79.68 | 90.75 (+0.60) |
| DTS-task | 74.15 | 76.85 | 97.92 | 99.66 | 97.00 | 98.34 | 99.63 | 79.03 | 90.32 |
| DTS-diff | 74.15 | 76.87 | 97.98 | 99.78 | 97.36 | 98.63 | 99.67 | 78.78 | 90.40 (+0.08) |

Table 16: Comparison of performance (%) between task vector (-task) and difference vector (-diff) applications to baseline methods for merging ViT-L/14 models on eight tasks. (The values) represent the performance gains.

| Method | SUN397 | Cars | RESISC45 | EuroSAT | SVHN | GTSRB | MNIST | DTD | Avg. |
|---|---|---|---|---|---|---|---|---|---|
| Individual | 81.72 | 92.39 | 98.85 | 99.88 | 98.11 | 99.24 | 99.69 | 84.15 | 94.25 |
| Weight Averaging (Wortsman et al., 2022) | 72.19 | 81.42 | 82.55 | 91.93 | 78.08 | 70.76 | 97.14 | 62.95 | 79.63 |
| Task-Arithmetic (Ilharco et al., 2023)-task | 73.92 | 82.13 | 87.64 | 92.82 | 87.91 | 86.77 | 98.94 | 65.64 | 84.47 |
| Task-Arithmetic (Ilharco et al., 2023)-diff | 72.39 | 82.54 | 82.83 | 92.42 | 78.23 | 70.65 | 97.01 | 62.55 | 79.83 (-4.64) |
| Ties-Merging (Yadav et al., 2023)-task | 74.74 | 84.50 | 89.00 | 94.18 | 85.66 | 82.07 | 98.65 | 67.71 | 84.56 |
| Ties-Merging (Yadav et al., 2023)-diff | 73.11 | 81.92 | 83.25 | 90.28 | 77.31 | 66.41 | 96.82 | 62.93 | 79.00 (-5.56) |
| EMR-Merging (Huang et al., 2024)-task | 80.47 | 90.71 | 98.55 | 99.54 | 97.94 | 99.10 | 99.69 | 82.71 | 93.59 |
| EMR-Merging (Huang et al., 2024)-diff | 81.78 | 92.31 | 98.85 | 99.90 | 98.12 | 99.23 | 99.75 | 83.67 | 94.20 (+0.61) |
| T-Switch (Qi et al., 2025)-task | 81.84 | 92.38 | 98.89 | 99.74 | 98.03 | 99.08 | 99.63 | 83.72 | 94.16 |
| T-Switch (Qi et al., 2025)-diff | 81.95 | 92.69 | 98.92 | 99.9 | 98.13 | 99.23 | 99.73 | 84.57 | 94.39 (+0.23) |
| DTS-task | 81.82 | 91.93 | 98.82 | 99.90 | 98.05 | 99.21 | 99.77 | 83.93 | 94.18 |
| DTS-diff | 81.75 | 92.09 | 98.89 | 99.88 | 99.27 | 99.25 | 99.72 | 84.15 | 94.24 (+0.06) |

Table 17: Comparison of performance (%) between task vector (-task) and difference vector (-diff) applications to baseline methods for merging RoBERTa models on eight tasks. (The values) represent the performance gains.

| Method | CoLA | SST2 | MRPC | STSB | QQP | MNLI | QNLI | RTE | Avg. |
|---|---|---|---|---|---|---|---|---|---|
| Individual | 60.18 | 94.04 | 89.22 | 90.63 | 91.41 | 87.20 | 92.71 | 79.06 | 85.56 |
| Weight Averaging (Wortsman et al., 2022) | 13.96 | 64.11 | 69.36 | 31.84 | 75.36 | 42.19 | 58.70 | 55.23 | 51.34 |
| Task-Arithmetic (Ilharco et al., 2023)-task | 18.78 | 85.89 | 79.90 | 74.03 | 83.78 | 59.08 | 69.67 | 62.09 | 66.65 |
| Task-Arithmetic (Ilharco et al., 2023)-diff | 9.28 | 79.47 | 77.70 | 31.80 | 79.41 | 40.03 | 72.36 | 61.01 | 56.38 (-10.27) |
| Ties-Merging (Yadav et al., 2023)-task | 20.48 | 84.40 | 81.13 | 58.19 | 85.70 | 64.65 | 74.81 | 42.96 | 64.04 |
| Ties-Merging (Yadav et al., 2023)-diff | 33.16 | 80.28 | 73.53 | 10.85 | 81.28 | 49.09 | 65.60 | 57.04 | 56.35 (-7.69) |
| EMR-Merging (Huang et al., 2024)-task | 39.96 | 93.35 | 86.27 | 82.73 | 89.72 | 85.45 | 89.57 | 74.37 | 80.18 |
| EMR-Merging (Huang et al., 2024)-diff | 51.70 | 93.46 | 89.22 | 88.17 | 91.19 | 87.13 | 92.24 | 75.45 | 83.57 (+3.39) |
| T-Switch (Qi et al., 2025)-task | 53.12 | 94.04 | 89.22 | 90.15 | 91.16 | 87.08 | 92.57 | 77.26 | 84.33 |
| T-Switch (Qi et al., 2025)-diff | 50.76 | 93.81 | 89.22 | 89.35 | 91.17 | 87.34 | 92.62 | 76.17 | 83.80 (-0.53) |
| DTS-task | 59.33 | 93.66 | 89.68 | 90.59 | 90.55 | 86.50 | 91.80 | 77.26 | 84.93 |
| DTS-diff | 59.66 | 93.69 | 89.71 | 90.62 | 90.80 | 87.09 | 92.11 | 76.17 | 84.98 (+0.05) |

Table 18: Comparison of performance (%) between task vector (-task) and difference vector (-diff) applications to baseline methods for merging GPT-2 models on eight tasks. (The values) represent the performance gains.

| Method | CoLA | MNLI | MRPC | QNLI | QQP | RTE | SST-2 | Avg. |
|---|---|---|---|---|---|---|---|---|
| Individual | 76.80 | 81.99 | 80.39 | 88.27 | 89.64 | 65.34 | 91.17 | 81.94 |
| Weight-Averaging (Wortsman et al., 2022) | 55.03 | 55.16 | 51.21 | 57.65 | 76.71 | 44.76 | 52.53 | 56.15 |
| Task-Arithmetic (Ilharco et al., 2023)-task | 68.71 | 68.64 | 69.56 | 70.42 | 81.83 | 47.21 | 83.62 | 70.00 |
| Task-Arithmetic (Ilharco et al., 2023)-diff | 55.03 | 59.24 | 50.98 | 57.60 | 76.69 | 44.76 | 52.52 | 56.69 (-13.31) |
| Ties-Merging (Yadav et al., 2023)-task | 68.48 | 71.46 | 68.49 | 69.56 | 82.57 | 47.68 | 81.82 | 70.01 |
| Ties-Merging (Yadav et al., 2023)-diff | 59.82 | 66.04 | 54.16 | 53.77 | 80.23 | 48.01 | 50.91 | 58.99 (-11.02) |
| EMR-Merging (Huang et al., 2024)-task | 72.77 | 81.08 | 79.16 | 84.84 | 88.11 | 66.43 | 90.25 | 80.38 |
| EMR-Merging (Huang et al., 2024)-diff | 74.01 | 81.80 | 80.14 | 88.33 | 89.47 | 67.14 | 90.82 | 81.67 (+1.29) |
| T-Switch (Qi et al., 2025)-task | 76.27 | 81.87 | 79.90 | 88.57 | 88.54 | 63.17 | 90.82 | 81.31 |
| T-Switch (Qi et al., 2025)-diff | 75.83 | 82.19 | 80.88 | 88.59 | 89.59 | 67.50 | 91.16 | 82.28 (+0.97) |
| DTS-task | 76.98 | 81.72 | 79.94 | 87.99 | 89.20 | 65.34 | 91.16 | 81.76 |
| DTS-diff | 76.69 | 81.58 | 80.63 | 88.24 | 89.46 | 64.98 | 91.05 | 81.80 (+0.04) |

## D.5 EXTENDING DIFFERENCE VECTOR TO SCENARIOS WHERE THE PRE-TRAINED MODEL IS INACCESSIBLE

Most prior model merging methods rely on task vectors, which require access to both pre-trained model parameters and fine-tuned model parameters. However, in scenarios where the pre-trained model is inaccessible—a more realistic and common setting—these methods become inapplicable. In contrast, this paper introduces the *difference vector*, which naturally extends to such cases. Specifically, we first obtain the merged model $\theta_m$ via simple weight averaging over $N$ fine-tuned models $\{\theta_n\}_{n=1}^{N}$, and then compute the difference vector as $d_n = \theta_n - \theta_m$. When the pre-trained model is inaccessible, the difference vector serves as a practical substitute for the task vector used in previous methods.

As shown in Tables 15, 16, 17, and 18, applying the difference vector to existing methods consistently yields performance that significantly surpasses simple weight averaging. Moreover, we compare difference vectors with task vectors and find that for conventional model merging methods, difference vectors result in slightly lower performance than task vectors, but still outperform naive averaging. In contrast, personalized model merging methods benefit from using difference vectors, often achieving higher accuracy than when using task vectors. Notably, for certain backbones, personalized models merged using difference vectors even surpass the performance of individually fine-tuned models. These results suggest that, compared to task vectors, the proposed difference vector provides a more accurate and transferable representation of task-specific information, particularly in the absence of the pre-trained model.

# E THE USE OF LARGE LANGUAGE MODELS

We use large language models (LLMs) solely for polishing the writing. It is important to note that the LLMs were not involved in the ideation, research methodology, or experimental design.

