# OpenReview forum: "Stay Unique, Stay Efficient: Preserving Model Personality in Multi-Task Merging"
_ICLR.cc/2026/Conference — ICLR 2026 Conference Withdrawn Submission_

### Official Review · Reviewer_irAB · 2025-10-24

**Soundness:** 2
**Presentation:** 2
**Contribution:** 1
**Rating:** 2
**Confidence:** 5

**Summary:**

The paper studies task-vector-based model merging, providing a multi-task model by aggregating the differences between task-specific finetunings and their common base model (task vectors). In particular, the work proposes DTS, a 3-step pipeline consisting of i) “Decomposition”, a layer-wise low-rank approximation of the differences, ii) “Thresholding”, a bit-masking step followed by a partitioning in 4 groups to reduce the storage requirements, and iii) a “Scaling” step that ensures the new computed singular vectors are re-scaled to the original unitary norm. The approach is tested over ViT-B/16 and ViT-L/14 on a common 8-task vision benchmark, and using RoBERTa and GPT2 on the GLUE benchmark.

**Strengths:**

- The singular-vector-level thresholding is methodologically novel and might be the main strength of the method. The 1% extra storage requirement is impressive, and is obtained by leveraging both low-rank approximation and quantization.
- The method is data-free: there is no training, tuning or test-time adaptation involved. While these are often observed in the related literature, I strongly advise against using task-specific data to prevent limiting the practical adoption of the method.

**Weaknesses:**

- The method requires task information at inference time which is not obtained by routing. This makes the comparison with standard merging techniques unfair, as the model is expecting not only the classification head to be known, but also which task-specific parameters to use. While this assumption is present in some of the literature, it severely restricts the usefulness of the approach, and is often unrealistic in practice.
- Lack of motivation. The performance degradation on similar tasks is not a surprising phenomenon per se, and it is not clear to me why it would motivate the proposed approach. The choice of SVHN as a dataset over which to compute the similarity wrt the other tasks is also arbitrary.
- Outdated baselines. The paper does not compare against the more recent structured merging methods such as Iso-C [1], TSV [2] and KNoTs [4].
- Lack of novelty. The paper presents limited methodological novelty: its first and main step, taking the SVD of a task vector, is already widely explored in the field [1, 2, 3, 4, 5]; these papers are also not properly cited and discussed. The subsequent steps seem incremental and should be compared with existing literature: while different from a methological point of view, the thresholding step has a similar goal as the quantization performed in [6], and it is not clear whether it should be preferred to this one. Considering the “difference vector” with respect to the merged model seems arbitrary and provides very marginal gains (ranging from +0.04% to 0.08%). Overall, the paper presents no novel insights.
- Incomplete empirical evidence. The work does not consider the commonly used 14- and 20-tasks benchmark proposed in [7] and commonly used in most subsequent works [1, 2, 3, 4], with the considered 8-tasks benchmark being fairly saturated. Also lacking the results for the ViT-B/16 baseline.

While lacking a solid motivation is not by itself reason enough to reject the paper, the overall lack of novelty and shaky experimental evidence makes the work incremental and its contribution uncertain. Having carefully considered both the strengths and weaknesses, I am inclined to reject the paper at this time.

[1] Gargiulo, Antonio Andrea, et al. "Task singular vectors: Reducing task interference in model merging." *Proceedings of the Computer Vision and Pattern Recognition Conference*. 2025.

[2] Marczak, Daniel, et al. "No Task Left Behind: Isotropic Model Merging with Common and Task-Specific Subspaces." *Forty-second International Conference on Machine Learning*.

[3] Choi, Jiho, et al. "Revisiting weight averaging for model merging." *arXiv preprint arXiv:2412.12153* (2024).

[4] Stoica, George, et al. "Model merging with SVD to tie the Knots." *The Thirteenth International Conference on Learning Representations*.

[5] Lu, Zhenyi, et al. "Twin-merging: Dynamic integration of modular expertise in model merging." *Advances in Neural Information Processing Systems* 37 (2024): 78905-78935.

[6] Kim, Youngeun, et al. "Task Vector Quantization for Memory-Efficient Model Merging." *CoRR* (2025).

[7] Wang, Ke, et al. "Localizing task information for improved model merging and compression." *Proceedings of the 41st International Conference on Machine Learning*. 2024.

**Questions:**

- The method requires reconstructing the merged model on the fly for each task. Assuming that each sample has a different task, what is the required added latency?
- It is not entirely clear to me how to derive the 1% extra storage cost. Is this dependent on the model size? depth? number of tasks?
- The 4 in the 4-group quantization seems arbitrary. How was this chosen? were 2 and 8 groups tried?

---

> ### Author Response · Authors · 2025-11-14
>
> Thank you for your time and for recognizing our novel singular-vector-level thresholding and the data-free method. Below, we provide point-by-point responses to each concern and question raised.
>
> ---
> >**Q1:** The method requires task information at inference time, which is not obtained by routing.
>
> **A1:** Requiring the task ID is standard in prior basic model merging and personalized mdoel merging works[1,2,3,4], as it is necessary to select the correct label space and task-specific parameters. Our method follows the same assumption as existing approaches, and this requirement does not limit practical usability in typical multi-task deployment settings.
>
> [1] Editing models with task arithmetic.
>
> [2] Emr-merging: Tuning-free high-performance model merging.
>
> [3] Twin-merging: Dynamic integration of modular expertise in model merging.
>
> [4] Less is more: Efficient model merging with binary task switch.
>
> ---
>
> >**Q2:** Lack of motivation. The performance degradation on similar tasks is not a surprising phenomenon per se, and it is not clear to me why it would motivate the proposed approach. The choice of SVHN as a dataset over which to compute the similarity wrt the other tasks is also arbitrary.
>
> **A2:**
> While degradation across tasks is expected in general, our key observation is that substantial degradation still occurs even between tasks that are extremely similar. This indicates that task interference is stronger and more persistent than commonly assumed, which directly motivates preserving task-specific information rather than mixing parameters.
>
> SVHN was used only as one visual example; we deliberately repeated the same analysis on CoLA for NLP, obtaining consistent results. This demonstrates that the phenomenon is not tied to SVHN and supports the general motivation of our method.
>
> ---
>
> >**Q3:** Outdated baselines. The paper does not compare against the more recent structured merging methods such as Iso-C [1] and TSV [2].
>
> **A3:**
> We would like to clarify that our experiments already include recent structured merging methods, such as T-Switch (CVPR 2025), which is one of the latest and strongest baselines in this line of work.
> Additionally, we compare our method with Iso-C[1] and TSV[2] using ViT-B/32 and ViT-L/14 as backbones on eight tasks, reporting the average accuracy as shown below.
>
> |Method| ViT-B\32| ViT-L\14|
> |-|:-:|:-:|
> |Individual|93.05|94.25|
> |Iso-C|85.86| 89.01|
> |TSV|86.31|90.62|
> |DTS-T|92.37|94.18|
> |DTS-D|**92.41**| **94.24**|
> |DTS-T*|91.99|94.16|
> |DTS-D*|92.08|94.14|
>
> We found that DTS consistently outperforms them. Due to space constraints, these results were not included in the current draft, but we will add the full comparisons in the final version.
>
> [1] Task singular vectors: Reducing task interference in model merging. CVPR 2025
>
> [2] No Task Left Behind: Isotropic Model Merging with Common and Task-Specific Subspaces. CIML2025
>
> ---
>
> >**Q4:** The paper presents limited methodological novelty: its first and main step, taking the SVD of a task vector, is already widely explored in the field [1, 2, 3, 4, 5]. The subsequent steps seem incremental and should be compared with existing literature: while different from a methodological point of view, the thresholding step has a similar goal as the quantization performed in [6], and it is not clear whether it should be preferred to this one. Considering the “difference vector” with respect to the merged model seems arbitrary and provides very marginal gains.
>
> **A4:**
> We would like to clarify that our method differs substantially from prior SVD-based merging works.
> 1. **Beyond existing SVD-based approaches.** Prior works apply SVD to full parameter matrices or use singular vectors to study interference.
> In contrast, DTS performs rank-restricted SVD on task-specific deviations and stores only the top-r singular components, achieving ∼1% per-task storage—significantly more compact than all prior SVD-based merging methods. None of the cited works introduce a low-rank, memory-bounded representation tailored for personalized reconstruction.
>
> 2. **Fine-grained thresholding beyond existing quantization.** Compared with binary or coarse quantization methods, DTS introduces a four-group thresholding with group-wise scaling, providing a finer approximation while keeping memory extremely low. Ablation studies (Table 12) show that this design yields clear performance gains over standard quantization.
>
> 3. **Difference vector addresses the realistic “no pretrained model available” setting.** The difference vector is not arbitrary—it enables personalized merging when the pretrained model cannot be accessed, a scenario where task vectors cannot be used. Experiments show that DTS-D achieves performance on par with DTS-T, demonstrating that the proposed representation is both practical and effective.

---

> ### Author Response · Authors · 2025-11-14
>
> >**Q5:** Incomplete empirical evidence. The work does not consider the commonly used 14- and 20-tasks benchmark proposed in [7] and commonly used in most subsequent works [1, 2, 3, 4], with the considered 8-tasks benchmark being fairly saturated. Also lacking the results for the ViT-B/16 baseline.
>
> **A5:** Our method stores only a small task-specific deviation (~1% per task), and therefore its scalability is not sensitive to the number of tasks.
>
> We additionally used ViT-B/16 as the backbone model, fine-tuned on 30 datasets including MNIST, CIFAR-10, Food-101, Kvasir-v2, Cars, EuroSAT, Intel Images, and more. Full details are provided in [3].
>
> We applied both baseline and DTS merging methods across all 30 tasks. The average accuracy is shown below: (*The best result is highlighted in **bold**.*)
>
> |Method|Avg. (%)|
> |-|:-:|
> |Individual|93.05|
> |Weight Averaging|42.54|
> |Task Arithmetic|48.89|
> |Ties-Merging|37.53|
> |EMR-Merging|89.54|
> |T-switch|91.96|
> |DTS-T |92.37|
> |DTS-D |**92.41**|
> |DTS-T* |91.99|
> |DTS-D* |92.08|
>
>
> We observe that DTS consistently outperforms baseline methods and  maintains performance close to individual models, demonstrating strong scalability and robustness in large-scale, heterogeneous task settings.
>
>
> ---
>
>
> >**Q6:** The method requires reconstructing the merged model on the fly for each task. Assuming that each sample has a different task, what is the required added latency?
>
> **A6:**
> The reconstruction of ($\hat{\theta}_u$) is performed **only once per task at inference**, not at every forward pass. This computation involves a single low-rank SVD reconstruction with small r, whose FLOPs are negligible compared to running the full backbone. Therefore, the additional latency introduced by DTS during inference is minimal and does not affect practical deployment.
>
>
> ---
>
> >**Q7:** It is not entirely clear to me how to derive the 1% extra storage cost. Is this dependent on the model size? depth? number of tasks?
>
>
> **A7:**
> Thank you for pointing this out. The reported “1% extra storage” refers to the **per-task additional memory rate (AMR)**, defined as
> $$
> \mathrm{AMR} =
> \frac{ \text{memory for task-specific information per task}}
> { \text{memory of the full model}}.
> $$
> In other words, we first compute the storage required for the backbone parameters, then the storage required for the DTS task-specific information of a single task (low-rank SVD components + masks + scaling factors), and take their ratio. This quantity is **per task** and is independent of the number of tasks. It only depends on the backbone architecture through its parameter count, and we choose the sparsity factor ( r ) to keep this ratio ≈1% across different models.
>
>
> ---
>
> >**Q8:** The 4 in the 4-group quantization seems arbitrary. How was this chosen? were 2 and 8 groups tried?
>
> **A8:**
> The choice of 4 groups was driven by the trade-off between accuracy and storage. As shown in Table 12 of our supplementary material, using 2 groups (i.e., simple binarization) leads to a large performance drop, indicating that the approximation becomes too coarse. Increasing to 8 groups provides only very marginal accuracy improvements, while the storage cost grows significantly because more masks and scaling factors must be stored.
>
> In contrast, 4 groups achieve accuracy very close to the individually fine-tuned upper bound while keeping memory extremely low, making it the most effective and efficient choice.

---

### Official Review · Reviewer_zWHb · 2025-10-29

**Soundness:** 2
**Presentation:** 3
**Contribution:** 2
**Rating:** 4
**Confidence:** 4

**Summary:**

This paper proposes a DTS method for multi-task model merging. The proposed approach performs low-rank decomposition on task vectors or difference vectors and defines a mask matrix using a threshold. Finally, it reconstructs the expert model parameters through the low-rank components and the mask matrix. However, this approach seems inconsistent with the original goal of model merging, which is to enable a single model to perform all tasks without requiring task identifiers.

**Strengths:**

- This paper proposes DTS for multi-task model merging.
- The method is validated across multiple architectures and domains.
- The paper is well-structured, and the method is easy to follow.

**Weaknesses:**

- The proposed method restores expert models by adding low-rank and sparse components to the pretrained model. However, this approach seems somewhat redundant-one could directly perform low-rank decomposition or sparsification on the expert model itself and store those parameters, eliminating the need to retain the pretrained model, thus saving storage. What are the advantages of the proposed approach compared with directly saving compact expert models?
- In Equation (7), the method only utilizes information from the pretrained model and a single expert model, without leveraging cross-task knowledge transfer, which seems inconsistent with the fundamental goals of multi-task learning or model merging.
- The proposed method requires the task ID during inference, which significantly limits its practical usability.

**Questions:**

Refer to the Weaknesses section

---

> ### Author Response · Authors · 2025-11-14
>
> Thank you for your thorough and thoughtful review. We appreciate your appreciation of the method, soundness and extensibility of our framework.
> We also thank you for raising important concerns, which we address in detail below.
>
>
> ---
>
> >**Q1:** The proposed method restores expert models by adding low-rank and sparse components to the pretrained model. However, this approach seems somewhat redundant-one could directly perform low-rank decomposition or sparsification on the expert model itself and store those parameters, eliminating the need to retain the pretrained model, thus saving storage. What are the advantages of the proposed approach compared with directly saving compact expert models?
>
> **A1:**
> While it is possible to compress or sparsify the full expert model, this typically leads to significant accuracy degradation, because the expert model parameters are generally dense [1].
>
> In contrast, prior works have shown that task vectors are naturally sparse and low-rank [1,2], meaning that compressing them has minimal impact on performance. DTS leverages this property by compressing only the task-specific deviation, rather than the entire model, allowing us to achieve much smaller per-task storage while preserving accuracy.
>
> [1] Task Vector Quantization for Memory-Efficient Model Merging. ICCV2025
>
> [2] Model merging with SVD to tie the Knots. ICLR2025
>
>
> ---
>
> >**Q2:** In Equation (7), the method only utilizes information from the pretrained model and a single expert model, without leveraging cross-task knowledge transfer, which seems inconsistent with the fundamental goals of multi-task learning or model merging.
>
> **A2:**
> Our experiments (Fig. 1) show that even highly similar tasks exhibit strong parameter conflicts when merged, so directly mixing task vectors harms performance. Therefore, Eq. (7) intentionally reconstructs each model using only its own task-specific deviation to avoid interference. Cross-task knowledge transfer is introduced in our unseen-task setting (Sec. 3.3), where task-specific information from multiple seen tasks is adaptively fused.
>
>
>
> ---
>
> >**Q3:** The proposed method requires the task ID during inference, which significantly limits its practical usability.
>
> **A3:**
> Requiring the task ID is standard in prior basic model merging and personalized mdoel merging works[1,2,3,4], as it is necessary to select the correct label space and task-specific parameters. Our method follows the same assumption as existing approaches, and this requirement does not limit practical usability in typical multi-task deployment settings.
>
> [1] Editing models with task arithmetic.
>
> [2] Emr-merging: Tuning-free high-performance model merging.
>
> [3] Twin-merging: Dynamic integration of modular expertise in model merging.
>
> [4] Less is more: Efficient model merging with binary task switch.

---

### Official Review · Reviewer_VamT · 2025-11-01

**Soundness:** 3
**Presentation:** 2
**Contribution:** 3
**Rating:** 6
**Confidence:** 3

**Summary:**

This paper proposes Decomposition, Thresholding, and Scaling (DTS), a personalized model merging method designed to improve storage efficiency. The DTS method first applies Singular Value Decomposition (SVD) to a task's parameter matrix. Then, it further compresses by separating left or right singular matrix (U/V) into four groups of weights: large-positive, small-positive, large-negative, and small-negative, based on signs and the median values. Each of these four groups is then represented by a binary mask and a scaling factor. This allows for a reconstruction of the task-specific vector from minimal storage (binary masks + scalars). Experiments across vision and NLP benchmarks show that DTS achieves a significantly better performance/storage trade-off than the baselines

**Strengths:**

1. Offer better trade-off between performance and storage efficiency than the baselines.
2. The presentation of the methods is clear and easy to follow.
3. Introduce the difference vector which make the method practical for the case when the pretrained is inaccessible.
4. Investigation on unseen tasks is appreciated especially for real world uses.

**Weaknesses:**

1. The authors do not evaluate computational overhead at inference time. Basic merging methods produce a single model, while personalized methods like DTS require a reconstruction of each task. Therefore, DTS should have higher computational overhead than basic merging and it would be valuable to evaluate this aspect too.
2. Some missing details (see questions)

**Questions:**

1. How is the decomposition (Eq. 2) applied to the layers that have more than 2 dimensions such as a convolutional layer?
2. In an ideal case, the Eq. 8 would be equivalent to the weight averaging of the finetuned models. But is it possible to use other basic model merging at this step such as TIES-Merging or DARE?
3. The thresholding strategy splits weights into 4 groups (pos/neg, large/small). Is it possible to apply this recursively? For example, could the $g_m^+$ group be further split into two sub-groups (e.g., $g_m^{++}$ and $g_m^{+-}$) to create 8 total groups? Would this provide a better trade-off point, or do the storage costs (more masks and scaling factors) grow too quickly for the marginal performance gain?
4. The details regarding the text encoder for the unseen task setup seems to be missing. How sensitive is the generalization performance (Section 4.2) to the choice of the text encoder?
5. The difference vector $d_n = \theta_n - \theta_m$ (Appendix D.5) is proposed as a solution for when the pretrained weight is inaccessible, using $\theta_m$ as an average of fine-tuned models. This seems to allow us to use the fine-tuned models that derive from different pretrained weights. Would this be possible?


Writing
1. The $\odot$ symbol in Equation 5 is never explicitly defined. While it can be inferred as element-wise multiplication (Hadamard product), this should be stated for clarity.

---

> ### Author Response · Authors · 2025-11-14
>
> We thank the reviewer for their valuable feedback and for acknowledging the novelty and thoroughness of our experiments.
> Below, we provide detailed responses to each of the concerns and questions raised.
>
> ---
> >**Q1:** The authors do not evaluate computational overhead at inference time.
>
> **A1:**
> This computation of reconstruction involves a single low-rank SVD reconstruction with small r, whose FLOPs are negligible compared to running the full backbone. Therefore, the additional latency introduced by DTS during inference is minimal and does not affect practical deployment.
>
>
>
> ---
>
>
> >**Q2:**
> How is the decomposition (Eq. 2) applied to the layers that have more than 2 dimensions such as a convolutional layer?
>
> **A2:**
> For layers whose parameters have more than two dimensions (e.g., convolutional kernels), we reshape the tensor into a 2D matrix, apply the decomposition in Eq. (2) and the DTS approximation on this reshaped matrix, and then reshape the reconstructed matrix back to the original tensor shape.
>
>
>
> ---
>
>
> >**Q3:**
> In an ideal case, the Eq. 8 would be equivalent to the weight averaging of the finetuned models. But is it possible to use other basic model merging at this step such as TIES-Merging or DARE?
>
> **A3:**
> Yes, it is indeed possible to integrate Eq. 8 with other basic merging methods such as TIES-Merging or DARE.
>
>
> ---
>
> >**Q4:**
> The thresholding strategy splits weights into 4 groups (pos/neg, large/small). Is it possible to apply this recursively? For example, could the group be further split into two sub-groups (e.g., $g^{++}$ and $g^{+-}_m$ ) to create 8 total groups? Would this provide a better trade-off point, or do the storage costs (more masks and scaling factors) grow too quickly for the marginal performance gain?
>
> **A4:**
> While recursive splitting is possible in principle, we find it unnecessary in practice. Using 4 groups already brings DTS extremely close to the performance of the individually fine-tuned models (the upper bound), leaving very limited room for further improvement. Doubling the number of groups would substantially increase storage due to additional masks and scaling factors, yet the resulting accuracy gains are negligible. Thus, expanding to 8 groups provides minimal benefit at a much higher cost and does not represent a favorable trade-off.
>
>
> ---
>
> >**Q5:**
> The details regarding the text encoder for the unseen task setup seems to be missing. How sensitive is the generalization performance (Section 4.2) to the choice of the text encoder?
>
> **A5:**
> Thank you for the question. For unseen task generalization, we follow the standard CLIP setting and use the same text encoder paired with the vision encoder (e.g., CLIP’s text transformer for ViT-B/32 and ViT-L/14). This ensures that the semantic similarity is computed in the same multimodal embedding space as the backbone.
> Moreover, our method is not sensitive to the specific choice of the text encoder, because the adaptive weights rely only on relative cosine similarities between task descriptions as shown in Eq. 8 in our main paper.
>
>
>
> ---
>
>
> >**Q6:**
> The difference vector (as shown in Appendix D.5) is proposed as a solution for when the pretrained weight is inaccessible, using as an average of fine-tuned models. This seems to allow us to use the fine-tuned models that derive from different pretrained weights. Would this be possible?
>
> **A6:**
> Yes, this is indeed possible. One advantage of the proposed difference vector is that it does not rely on access to the original pretrained weights. As long as a merged reference model $θ_m$ can be obtained (e.g., via averaging), the difference vector $d_n = θ_n− θ_m$ remains well-defined. This makes the method applicable even when the fine-tuned models originate from different pretrained checkpoints—a scenario where task vectors cannot be used. We will clarify this benefit in the final version.

---

### Official Review · Reviewer_BKB6 · 2025-11-01

**Soundness:** 2
**Presentation:** 2
**Contribution:** 2
**Rating:** 4
**Confidence:** 3

**Summary:**

This paper addresses the performance degradation commonly observed in model merging, especially when combining models fine-tuned on similar tasks. The authors propose DTS (Decomposition, Thresholding, and Scaling)—a personalized, approximation-based merging method designed to preserve task-specific information with minimal storage overhead. DTS performs singular value decomposition (SVD) on task-specific information (e.g., task or difference vectors), retains top-r singular components, and applies a novel thresholding and scaling strategy to compress and reconstruct parameters efficiently. A data-free variant further generalizes to unseen tasks by weighting stored components according to semantic similarity between task characteristics. Experiments across visual, language understanding, and generation tasks demonstrate that DTS outperforms strong baselines such as T-Switch and EMR-Merging while requiring ≈ 1% additional storage per task.

**Strengths:**

1)Comprehensive empirical validation: The paper conducts large-scale experiments across multiple domains and backbones, including ViT-B/32, RoBERTa, GPT-2, and Qwen-14B (Sec. 4.1). Results consistently show that DTS outperforms both basic and personalized baselines while maintaining very low storage overhead. For example, DTS-D achieves 90.40% average accuracy on vision tasks with only 3.68% extra memory (Table 1), while achieving near-individual model performance on RoBERTa with 0.88% overhead (Table 2). This extensive coverage and consistent trend strongly validate the proposed method’s generality and robustness.

2)Novel and effective compression mechanism: The decomposition thresholding scaling pipeline (Sec. 3.2; Fig. 3) is a distinctive and technically sound design that effectively preserves important task information under extreme compression. The use of four-value thresholding with learned scaling factors improves upon prior binary masking approaches. The ablation results (Table 12) show that removing scaling or thresholding leads to 2–5% accuracy drops, demonstrating that each step contributes meaningfully to preserving task personality.

3)Clarity, structure, and reproducibility: The paper is clearly written, logically structured, and easy to follow. The motivation in Fig. 1 and Sec. 1 is convincing, the method section (Sec. 3) is self-contained, and reproducibility is explicitly discussed (Sec. 7). The inclusion of code in the supplementary material further supports transparency and practical usability.

**Weaknesses:**

1)Lack of theoretical grounding for the method: Although DTS shows strong empirical performance, there is no theoretical analysis of its approximation quality or why the proposed thresholding and scaling preserve task identity. The manuscript does not provide error bounds for SVD truncation (Eq. 2) or theoretical justification for using four quantization groups (Sec. 3.2).

2)Inference cost ignored: Reconstructing $\hat{\theta}_u$ at every forward pass (Eq. (7)) adds matrix multiplications whose FLOPs and latency are neither analysed nor benchmarked (Sec. 3.2).

3)Hyper-parameter selection opaque: The claim “r is adaptively adjusted to keep storage <1 %” lacks an algorithm.

4)Incomplete fairness and statistical reporting: the experiments appear single-run with no mention of variance or confidence intervals.

**Questions:**

1) Figure 1 is drawn unclearly, and the overall explanation is ambiguous.

---

> ### Author Response · Authors · 2025-11-14
>
> Thank you for the insightful reviews and for recognizing our novel compression mechanism and the extensive experimental results.
>
> ---
> >**Q1:** Lack of theoretical grounding for the method: Although DTS shows strong empirical performance, there is no theoretical analysis of its approximation quality or why the proposed thresholding and scaling preserve task identity. The manuscript does not provide error bounds for SVD truncation (Eq. 2) or theoretical justification for using four quantization groups (Sec. 3.2).
>
> **A1:**
> The extensive results across multiple backbones already demonstrate that the proposed approximation reliably preserves task-specific information. Additionally, the ablation studies in the supplementary material clearly validate the necessity of both the thresholding and scaling components. We agree that providing formal error bounds for SVD truncation and a theoretical justification for the four-group quantization is an important direction, and we will include these analyses in future work.
>
>
>
> ---
>
> >**Q2:** Inference cost ignored: Reconstructing $\hat{\theta}_u$ at every forward pass adds matrix multiplications whose FLOPs and latency are neither analysed nor benchmarked.
>
> **A2:**
> The reconstruction of $\hat{\theta}_u$ is performed **only once per task at inference**, not at every forward pass. This computation involves a single low-rank SVD reconstruction with small r, whose FLOPs are negligible compared to running the full backbone. Therefore, the additional latency introduced by DTS during inference is minimal and does not affect practical deployment.
>
>
>
> ---
>
> >**Q3:**
> Hyper-parameter selection opaque: The claim “r is adaptively adjusted to keep storage <1 %” lacks an algorithm.
>
> **A3:**
> In our method, the sparsity coefficient r can be dynamically adjusted according to the storage budget. To ensure that the additional storage remains below 1%, r can be computed based on the ratio between the number of parameters before and after the SVD-based decomposition.
>
>
>
> ---
>
> >**Q4:**
> Incomplete fairness and statistical reporting: the experiments appear single-run with no mention of variance or confidence intervals.
>
> **A4:**
> Thank you for pointing this out. Our method does not involve any training or stochastic optimization—DTS is entirely training-free and deterministic. As a result, it produces the same merged model for a given set of fine-tuned models, and there is no inherent run-to-run variance to report. This is consistent with prior training-free merging methods, which also do not include variance or confidence intervals for the same reason.
>
>
>
> ---
>
>
> >**Q5:**
> Figure 1 is drawn unclearly, and the overall explanation is ambiguous.
>
> **A5:**
> Figure 1 illustrates our key motivation: even models fine-tuned on highly similar tasks still cause severe performance degradation when merged. For each reference task (SVHN in vision and CoLA in NLP), we pairwise merge its fine-tuned model with seven others and report the best (most similar task) and worst (most dissimilar task) results across merging methods.
> The figure shows that:
> 1. Even merging SVHN with MNIST — its closest task — results in a notable drop.
> 2. NLP tasks exhibit the same pattern with CoLA.
>
> These observations highlight that parameter conflicts exist even between semantically related tasks, reinforcing the necessity of preserving task-specific information, which motivates our DTS method.

---

### Note · Authors · 2025-11-14

I have read and agree with the venue's withdrawal policy on behalf of myself and my co-authors.